# Token-Efficient Change Detection in LLM APIs

**Timothée Chauvin** [* 1] **Clément Lalanne** [† 2 3] **Erwan Le Merrer** [1] **Jean-Michel Loubes** [4 3] **François Taïani** [1] **Gilles Tredan** [5]

## Abstract

Remote change detection in LLMs is a difficult problem. Existing methods are either too expensive for deployment at scale, or require initial white-box access to model weights or grey-box access to log probabilities. We aim to achieve both low cost and strict black-box operation, observing only output tokens. Our approach hinges on specific inputs we call Border Inputs, for which there exists more than one output top token. From a statistical perspective, optimal change detection depends on the model's Jacobian and the Fisher information of the output distribution. Analyzing these quantities in low-temperature regimes shows that Border Inputs enable powerful change detection tests. Building on this insight, we propose the Black-Box Border Input Tracking (B3IT) scheme. Extensive *in vivo* and *in vitro* experiments show that Border Inputs are easily found for the majority of tested endpoints, and achieve performance on par with the best available grey-box approaches. B3IT reduces costs by $30\times$ compared to existing methods, while operating in a strict black-box setting.

## 1. Introduction

LLM APIs are increasingly embedded in high-impact applications, such as software engineering (Zheng et al., 2025), yet users have no guarantee that the model behind an endpoint remains unchanged. Providers may modify models for safety updates, cost optimization, or infrastructure changes, often without notice. Quietly deploying quantized versions to reduce costs, for instance, could degrade performance on tasks users depend on. Such changes are not hypothetical: Grok on X suffered three incidents in 2025 where modified system prompts were deployed due to rogue employees or bad updates (Babuschkin, 2025; xAI, 2025a;b). Anthropic experienced infrastructure bugs in 2025 that degraded Claude's responses, which remained undetected for weeks despite affecting 16% of Claude Sonnet 4 requests at peak (Anthropic, 2025).

Several change-detection methods have been proposed (Gao et al., 2025; Bai et al., 2025; Gubri et al., 2024; Chauvin et al., 2026). However, they face a fundamental tension between access requirements and cost. White-box methods like ESF (Bai et al., 2025) and TRAP (Gubri et al., 2024) craft sensitive inputs using model internals (embeddings, gradient-based optimization), limiting their use to known models. Grey-box methods like LT (Chauvin et al., 2026) require log-probabilities, which are available only on a fraction of public endpoints. Fully black-box approaches like MET (Gao et al., 2025) avoid these requirements but demand many queries to the tracked model, making continuous monitoring prohibitively expensive.

We introduce B3IT (*Black-Box Border Input Tracking*), a change-detection method that operates in a strict black-box setting, observing only output tokens, while achieving accuracy and cost efficiency comparable to grey-box approaches. Our key insight, brought by a theoretical analysis, is that at low temperature, inputs where two tokens are nearly tied as the top prediction ("Border Inputs") become extremely sensitive detectors. In other words, even tiny model changes can dramatically alter their output distribution. We show theoretically that this sensitivity diverges as temperature tends to zero, exhibiting a phase transition between detectable and undetectable regimes.

B3IT exploits this phenomenon: it discovers Border Inputs through black-box sampling, then monitors for changes by comparing output distributions. Surprisingly, Border Inputs are easy to find on most production models. The result is a method that detects subtle changes at $1/30$th the cost of alternatives, enabling large-scale continuous and cost-effective API monitoring.

---

*First author, method and experiments. †First author, theory and analysis. [1]Université de Rennes, Inria, CNRS/IRISA (Rennes, France) [2]Univ Toulouse, INUC, UT2J, INSA Toulouse, TSE, CNRS, IMT, Toulouse, France [3]ANITI Toulouse [4]Equipe Regalia, Inria (Bordeaux, France) [5]LAAS, CNRS, (Toulouse, France). Correspondence to: Timothée Chauvin <https://tchauvin.com/contact>.

*Proceedings of the 43rd International Conference on Machine Learning*, Seoul, South Korea. PMLR 306, 2026. Copyright 2026 by the author(s).

**Contributions.** **(1)** We establish theoretical foundations for change detection in LLMs, revealing a phase transition at low temperature that enables high-sensitivity detection without model access. **(2)** We propose B3IT, a practical black-box method that identifies Border Inputs and uses them for efficient change detection. **(3)** We validate B3IT *in vitro* on TinyChange (Chauvin et al., 2026), achieving detection performance comparable with grey-box methods at $1/30$th the cost of the next best black-box baselines. **(4)** We demonstrate B3IT *in vivo* on 131 endpoints (78 models, 34 providers), showing broad applicability for continuous monitoring.

**Roadmap.** Section 2 describes our system model. Section 3 presents the theoretical analysis underpinning the B3IT method, which we describe in Section 4 and evaluate in Section 5.

**Code.** All our code is open-source at github.com/timothee-chauvin/token-efficient-change-detection-llm-apis.

## 2. System model

In this paper, we take the perspective of an unprivileged user facing a black-box API exposing an LLM $f_\theta$, where $\theta \in \Theta$ denotes the (unknown) model parameters. The user can only submit queries $(x, T)$ for some input $x \in \mathcal{X}$ and temperature $T \in \mathbb{R}^+$, and collect the answer $f_\theta(x, T)$.

**Change detection.** We consider a two-stage interaction: an *initialization stage* and a *detection stage*. Let $\theta_0$ denote the model parameters during initialization and $\theta_1$ during detection. The change detection problem consists in deciding whether $\theta_0 = \theta_1$, as in prior work (Gao et al., 2025; Chauvin et al., 2026). Formalized as a hypothesis test, let $H_0 := \theta_0 = \theta_1$ (the model hasn't changed) versus $H_1 := \theta_0 \neq \theta_1$ (the model has changed).

In this endeavor, we are interested in two key quantities:

- **Detector accuracy**: we want to minimize false positives (incorrectly detecting a change) and false negatives (missing actual changes). Formalizing a test as a (possibly randomized) function $\phi$ taking values in $[0, 1]$, we therefore seek to minimize $\mathbb{P}(\phi = 1|H_0) + \mathbb{P}(\phi = 0|H_1)$.

- **Cost**: for wide-scale monitoring of LLM APIs, detecting changes should induce a minimal cost. This is especially true of the detection stage, which is run repeatedly.

**Types of changes.** Deployed LLMs can undergo many kinds of modifications: fine-tuning (full or Low-Rank Adaptation (Hu et al., 2022)), quantization, model distillation, system prompt changes, or further Reinforcement Learning.

Infrastructure can also affect outputs: CUDA versions, GPU selection, compiler optimizations, or routing errors directing requests to misconfigured servers (Anthropic, 2025). The intensity of any such modification determines detection difficulty; our focus is on query-efficient detection, regardless of the change's origin or intensity.

For the theoretical analysis, we model parameter changes as $\theta_1 := \theta_0 + \epsilon h$ where $h \in \mathbb{S}^{q-1}$ is a unit direction and $\epsilon$ controls magnitude. This local perturbation model enables tractable analysis via the Local Asymptotic Normality framework (Section 3). In experiments, we directly evaluate on realistic updates (fine-tuning, pruning, etc.) across a range of change magnitudes, and on live LLM APIs.

**Single-token perspective.** Our analysis focuses on the first output token from a single query. While approaches such as MET (Gao et al., 2025) leverage multiple output tokens, inter-token dependencies complicate the analysis substantially. However, multiple queries from different prompts can be combined for improved performance, which we demonstrate experimentally.

## 3. Theoretical analysis and guidelines

Building on optimal change detection in the sense of Neyman–Pearson (Neyman & Pearson, 1933), we exhibit a new phase-transition phenomenon for LLMs (Theorem 3.3). At temperature $T \ll 1$, inputs where two or more tokens share the maximal logit become extremely sensitive to parameter changes: we call these *Border Inputs* (BIs). This result provides the theoretical foundation for B3IT, which we introduce in Section 4.

### 3.1. Rationale

Testing whether two LLMs differ reduces to testing whether they induce different probability distributions over tokens. We focus on the distribution of the first generated token and, for analytical tractability, assume independence across generations.

Since the token distribution is finite, the problem formally reduces to multinomial hypothesis testing, a classical setting with an extensive literature directly studying the problem, or related ones such as estimation (Balakrishnan & Wasserman, 2019; Berrett & Butucea, 2020; Cai et al., 2024; Aliakbarpour et al., 2024; Cressie & Read, 2018; Barron, 1989; Kim, 2019; Acharya et al., 2019; 2018; Kairouz et al., 2016; Acharya et al., 2017a;b; Louati, 2025). However, these generic results do not account for the structure of LLMs, or for the ability to design informative input prompts. We therefore analyze how prompt choice and sampling temperature shape sensitivity to model perturbations, ultimately identifying BIs as appealing test inputs.

### 3.2. The Local Asymptotic Normality Framework

Let $\mathbf{p} = (p_1, \ldots, p_{d-1}) \in (0,1)^{d-1}$ denote the reduced parametrization of the categorical output distribution, with $p_d := 1 - \sum_{i=1}^{d-1} p_i > 0$. The vector $\mathbf{p}$ models the output distribution over tokens, where $d$ is the vocabulary size. We adopt this reduced parametrization to isolate the free variables. Querying the model $n$ times on the same input yields i.i.d. samples $Y_1, \ldots, Y_n \sim \mathbf{p}$.

We define the reduced empirical frequency estimator $T_n := \hat{p} := (\hat{p}_1, \ldots, \hat{p}_{d-1}), \hat{p}_j := \frac{1}{n} \sum_{i=1}^{n} \mathbf{1}\{Y_i = j\}$ . This estimator is unbiased and satisfies $\mathrm{Cov}_{\mathbf{p}}(T_n) = \frac{1}{n} F(\mathbf{p})$, where $F(\mathbf{p}) := \mathrm{diag}(\mathbf{p}) - \mathbf{p}\mathbf{p}^T$ is the inverse of the Fisher information matrix of the distribution $\mathbf{p}$. Since $T_n$ is a sufficient statistic, it contains all the information needed to construct optimal tests.

Consider two sets of model parameters $\theta_0$ and $\theta_1$, inducing token distributions $\mathbf{p}_0$ and $\mathbf{p}_1$, respectively. By the Central Limit Theorem (see (van der Vaart, 1998), Example 2.18), the statistic $T_n$ admits the asymptotic Gaussian approximation $T_n \sim \mathcal{N}\left(\mathbf{p}_k, \frac{1}{n} F(\mathbf{p}_k)\right), k \in \{0,1\}$.

Let $\theta \in \Theta \subset \mathbb{R}^q$ denote the model parameters, let $x_{\text{test}}$ be the prompt used for testing, and define the reduced output map $g(\theta, x_{\text{test}}) := f(\theta, x_{\text{test}})_{1:(d-1)} \in (0,1)^{d-1}$ . We consider a perturbation of the parameters of the form $\theta \mapsto \theta + \epsilon h, h \in \mathbb{S}^{q-1}$, and write $\mathbf{p}_1 := g(\theta + \epsilon h, x_{\text{test}}), \mathbf{p}_0 := g(\theta, x_{\text{test}})$. Relating the difficulty of change detection to parameters arising in LLM inference requires particular care. In practice, the model landscape is highly complex, and the effect of parameter perturbations on likelihood ratios quickly becomes intractable. As a result, one is often forced to rely on standard multinomial testing, which does not reveal how inference-specific parameters, such as the prompt or the temperature, affect the difficulty of the test. To address this limitation, we focus on the *local* regime. Indeed, in a neighborhood of a reference parameter, even highly complex models can often be approximated through their Taylor expansion. This perspective allows us to derive a meaningful characterization of the difficulty of detecting small parameter changes and to connect it to control knobs relevant to LLM inference.

A further difficulty arises from the statistical nature of the problem. We do not observe the model directly, but only samples from its output distribution, so the sample size must be calibrated carefully to avoid degenerate regimes: with too many samples, the testing problem becomes essentially trivial but costly, whereas with too few, it is nearly impossible. Consequently, when considering a local asymptotic regime in which the parameters approach one another and the testing problem becomes intrinsically harder, the sample size must scale accordingly.

A meaningful theory emerges by considering small parame-

ter perturbations in the Local Asymptotic Normality (LAN) regime (see (van der Vaart, 1998), Chapters 7 and 16), which we now introduce. This framework is central in statistics, as it characterizes the regime in which hypothesis testing is neither trivial nor impossible.

Assuming $g(\theta, x_{\text{test}})$ is differentiable with respect to $\theta$, we consider local perturbations of the form $\epsilon_n = \frac{s}{\sqrt{n}}$ for $s \in \mathbb{R}$ fixed, and obtain the expansion $\mathbf{p}_n := \mathbf{p}_{\epsilon_n} = \mathbf{p}_0 + \epsilon_n J h + r_n$, where $J := \nabla_\theta g(\theta, x_{\text{test}})$ is the Jacobian of the reduced output distribution w.r.t. the model parameters, and $\sqrt{n}\|r_n\| \to 0$. This corresponds to a small parameter perturbation in direction $h$ at the critical rate that preserves nontrivial asymptotic effects.

Leveraging the perspective of this regime leads to the following result, which quantifies the asymptotic power of optimal tests.

**Theorem 3.1** (Optimal Tests in the LAN Regime)**.** *Let* $\alpha \in (0,1)$. *If* $(\phi_n)$ *is a sequence of tests such that, for every* $n$, $\phi_n$ *is the test with the lowest Type-II error among all tests with Type-I error at most* $\alpha$ *in testing* $\mathbf{p}_0$ *vs* $\mathbf{p}_n$, *then*

$$\text{Type-II}(\phi_n) \to \mathbb{P}\left(\mathcal{N}(0,1) \leq Q_\alpha - \sqrt{s^2 \mathrm{SNR}^2(h)}\right) \tag{1}$$

*where* $\mathrm{SNR}^2(h) := h^T (J^T F(\mathbf{p}_0)^{-1} J) h$, *and Type-II*$(\phi_n)$ *refers to the error of* $\phi_n$ *under* $\mathbf{p}_n$, *and where* $Q_\alpha$ *is the quantile of order* $1 - \alpha$ *of* $\mathcal{N}(0,1)$.

*Proof.* See Appendix B.2. □

This result is central to our theory, as it shows that the viability of the optimal (and thus any) test is entirely characterized by a single scalar quantity, $\mathrm{SNR}^2(h)$, which captures the intrinsic difficulty of the testing problem: the higher it is, the easier the test. It conveniently takes the form of a signal-to-noise ratio, where the signal is the change in intensity, and the noise comes from temperature sampling.

**Takeaway.** In the small-variation regime, optimal change detection is governed by three elements: the tampering direction $h$, the Jacobian of the model, and the Fisher information of the output token distribution, combined through the criterion $\mathrm{SNR}^2(h)$. The Jacobian controls the signal: a large Jacobian amplifies how parameter perturbations affect the output distribution. The Fisher information controls the noise: a well-conditioned (low-variance) output distribution reduces sampling uncertainty and improves detectability.

Balancing these effects may seem difficult in a black-box setting. We show, however, that tuning the sampling temperature provides concrete control over this trade-off.

### 3.3. Decomposition for LLMs: exploiting the last layer

A key observation underlying our theory is that, for LLMs, the expression of $\text{SNR}^2(h)$ can be further simplified by making the transformer architecture explicit.

Let $r(\theta, x_{\text{test}}) \in \mathbb{R}^m$ denote the representation produced by the model up to the last layer. The final layer is a linear head $(W, b)$ producing logits $z(\theta) = z(\theta, x_{\text{test}}) = Wr(\theta, x_{\text{test}}) + b \in \mathbb{R}^d$. Given a temperature $\tau > 0$ (denoted $T$ elsewhere in the paper; we switch to $\tau$ in this theoretical analysis to avoid collision with the test statistic $T_n$), the output token distribution is computed as the softmax of the logits: $p_i^{(\tau)}(\theta) = \frac{\exp(z_i(\theta)/\tau)}{\sum_{j=1}^d \exp(z_j(\theta)/\tau)}$, for $i = 1, \ldots, d$.

We will write $\mathbf{p}^{(\tau)}(\theta) \in \mathbb{R}^d$ for the full vector and $\mathbf{p}_{1:(d-1)}^{(\tau)}(\theta) \in \mathbb{R}^{d-1}$ for its reduced coordinates.

We define the full matrix $\Sigma(\mathbf{p}) := \text{diag}(\mathbf{p}) - \mathbf{p}\mathbf{p}^T \in \mathbb{R}^{d \times d}$, and recall the reduced matrix $F(\mathbf{p}_{1:(d-1)}) := \text{diag}(\mathbf{p}_{1:(d-1)}) - \mathbf{p}_{1:(d-1)}\mathbf{p}_{1:(d-1)}^T \in \mathbb{R}^{(d-1) \times (d-1)}$. We use these two different notations to better distinguish reduced coordinates from full coordinates.

We write the parameters as $\theta = (\theta_{\text{pre}}, W, b)$, where $\theta_{\text{pre}}$ collects all parameters before the head, so that $r(\theta, x_{\text{test}}) = r(\theta_{\text{pre}}, x_{\text{test}})$. We define $J_r(\theta_{\text{pre}}) := \frac{\partial r(\theta_{\text{pre}}, x_{\text{test}})}{\partial \theta_{\text{pre}}} \in \mathbb{R}^{m \times q_{\text{pre}}}$. Then the Jacobian of the logits w.r.t. the full parameter vector $J_z(\theta) \in \mathbb{R}^{d \times (q_{\text{pre}} + dm + d)}$ decomposes as

$$J_z(\theta) = \Big[ \underbrace{W J_r(\theta_{\text{pre}})}_{\in \mathbb{R}^{d \times q_{\text{pre}}}} \Big| \underbrace{r(\theta_{\text{pre}}, x_{\text{test}})^T \otimes I_d}_{\in \mathbb{R}^{d \times (dm)}} \Big| \underbrace{I_d}_{\in \mathbb{R}^{d \times d}} \Big].$$

Finally, let $J^{(\tau)}(\theta) := \frac{\partial \mathbf{p}_{1:(d-1)}^{(\tau)}(\theta)}{\partial \theta} \in \mathbb{R}^{(d-1) \times (q_{\text{pre}} + dm + d)}$ be the Jacobian of the reduced output map.

**Lemma 3.2** ($\text{SNR}^2(h)$ in LLMs). *For any $\tau > 0$,*

$$\text{SNR}^2(h) = \frac{1}{\tau^2} h^T \left( J_z(\theta)^T \Sigma\left(\mathbf{p}^{(\tau)}(\theta)\right) J_z(\theta) \right) h. \quad (2)$$

*Proof.* See Appendix B.3. □

Although purely technical, this result is crucial to our theory: it renders the analysis tractable for LLMs, with the simplification of the inner inverse arising from the structure of the softmax layer, and enables the derivation of our main result.

### 3.4. The Zero-Temperature Limit

We now have all the ingredients to study the low-temperature regime and present our main theoretical result. Let $z := z(\theta)$ and define the set of maximizers $\mathcal{M} := \{i \in \{1, \ldots, d\} : z_i = \max_{1 \leq j \leq d} z_j\}, k := |\mathcal{M}|$.

We further define $\Sigma_{\mathcal{M}}$ as the operator $\Sigma$ applied to the probability vector assigning uniform mass to $\mathcal{M}$ and zero mass elsewhere.

**Theorem 3.3** (Phase Transition). *Depending on the value of $k$,*

- *If $k = 1$, then $\text{SNR}^2(h) \to 0$ as $\tau \to 0$.*

- *If $k \geq 2$, and if $h^T \left( J_z^T \Sigma_{\mathcal{M}} J_z \right) h \neq 0$, then $\text{SNR}^2(h) \to +\infty$ as $\tau \to 0$.*

*Proof.* See Appendix B.4. □

*Remark* 3.4 (On the condition $h^T \left( J_z^T \Sigma_{\mathcal{M}} J_z \right) h \neq 0$). At first glance, the condition $h^T \left( J_z^T \Sigma_{\mathcal{M}} J_z \right) h \neq 0$ may appear restrictive. However, as discussed in Appendix B.5, it holds for almost every direction $h$. In particular, if $h$ is not degenerate and is, for instance, modeled as a continuous random perturbation of the parameters, this condition is satisfied with probability one.

**Takeaway.** At very low temperature, detectability exhibits a sharp dichotomy. If the output distribution collapses to a single token, then $\text{SNR}^2(h) \to 0$ and detection becomes nearly impossible: the distribution converges so rapidly to a Dirac mass that small parameter perturbations cannot alter the outcome. If at least two logits are tied, $\text{SNR}^2(h)$ diverges and detectability is maximal: in the presence of ties, arbitrarily small parameter changes can force the model to select a single mode, making unimodal versus multimodal behavior easy to test. Moreover, under the conditions of Theorem 3.3, this divergence is obtained regardless of the model's weights, enabling a black-box approach.

A natural strategy for change detection is therefore to select $x_{\text{test}}$ such that the logit distribution has at least two maximal entries. These inputs (termed **Border Inputs** or BIs) lie at the core of the change detection method presented next.

## 4. B3IT: Black-Box Border Input Tracking

Building upon Section 3, the B3IT method comprises two stages: $i)$ an initialization stage whose goal is to identify Border Inputs (BIs) and their support, and $ii)$ a detection stage whose goal is to detect a potential change of the target model compared to the initialization stage. As BIs leverage the low temperature phase transition, they exhibit high sensitivity to model changes: intuitively, if an input identified as a BI during initialization remains a BI during detection, then the model has likely not changed. We now describe both stages.

## 4.1. The B3IT Initialization Stage

The first step consists of finding BIs. While intuitively unlikely, in practice finding BIs turns out to be relatively easy on the majority of production models we tested, which allows B3IT to use a simple procedure: submit $n$ random inputs $m$ times at the lowest temperature $T$ (possibly $T = 0$), and retain as BIs the $\bar{n}$ inputs that produced more than one distinct output across their $m$ draws.

More precisely, the phase transition phenomenon explored in Theorem 3.3 states that at $T \approx 0$ temperature, given an input $x$, only two situations can arise: either $x$ is not a BI ($k = 1$), in which case it will always produce the same output token, or $x$ is a $BI$ ($k \geq 2$) and the output token is uniformly sampled among top tokens: a multinomial output distribution is observed.

The key parameter driving this stage is $m$, the number of samples collected per candidate BI $x$. A back-of-the-envelope analysis (see Appendix C) identifies $m = 3$ as an optimal value if less than $75\%$ of the candidates are BIs. This is the value we select in B3IT, which enables the discovery of BIs for less than $1,500$ requests in a majority of production APIs (See Section 5.2).

Let $x_{\text{test}}$ be an identified BI. To accurately estimate its initial distribution, it is sampled $n_1$ times. The complete initialization procedure is given as pseudocode in Algorithm 1.

## 4.2. The B3IT Detection Stage

The detection stage is to be triggered when a user wants to determine whether the target model has changed since the initialization stage. During detection, the temperature is set to its minimum (possibly $T = 0$) to operate in the optimal test regime.

In this regime, one can design simple tests with fully controlled nonasymptotic guarantees that are optimal for the practical use case considered here. As the output distribution concentrates on a (typically small) subset of tokens $\mathcal{M}$ attaining the maximal log-probability, each with probability $1/k$, all remaining tokens receive zero probability. Consequently, the observed generation behavior can be modeled as sampling from a uniform distribution supported on an unknown finite subset of the token vocabulary.

Formally, we model the reference $\theta_0$ and candidate $\theta_1$ models by two unknown supports $S_1, S_2 \subset \Omega := \{1, \ldots, d\}$ with corresponding output distributions $P = \text{Unif}(S_1)$ and $Q = \text{Unif}(S_2)$. We observe independent samples $X_1, \ldots, X_{n_1} \sim P$ and $Y_1, \ldots, Y_{n_2} \sim Q$, and aim to test whether the two models induce the same effective support, namely $H_0 : S_1 = S_2$ versus $H_1 : S_1 \neq S_2$.

In this setting, observing a token under one model that never appears under the other directly indicates a support mismatch. We therefore define the empirical supports $\hat{S}_1 := \{X_1, \ldots, X_{n_1}\}$ and $\hat{S}_2 := \{Y_1, \ldots, Y_{n_2}\}$, and the rejection event $\mathcal{R} := (\hat{S}_1 \setminus \hat{S}_2) \cup (\hat{S}_2 \setminus \hat{S}_1) \neq \emptyset$. In other words, B3IT rejects $H_0$ whenever $\mathcal{R}$ occurs. The full detection procedure is given as pseudocode in Algorithm 2.

## 4.3. Analysis: Type-I and Type-II Errors

In this limit regime, we provide nonasymptotic guarantees for both Type-I and Type-II errors. We begin by controlling the Type-I error.

**Theorem 4.1** (Type-I error). *Under* $H_0 : S_1 = S_2 =: S$ *with* $|S| = k$, $\mathbb{P}_{H_0}(\mathcal{R}) \leq k\left(1 - \frac{1}{k}\right)^{n_1} + k\left(1 - \frac{1}{k}\right)^{n_2}$ $\leq ke^{-n_1/k} + ke^{-n_2/k}$.

*Proof.* See Appendix B.6. □

We may also control the Type-II error as follows.

**Theorem 4.2** (Type-II error). *Let* $I := S_1 \cap S_2$, $k_1 := |S_1|$, $k_2 := |S_2|$ *and set* $p_1 := \frac{|I|}{k_1} \in [0, 1]$, $p_2 := \frac{|I|}{k_2} \in [0, 1]$. *Under* $H_1 : S_1 \neq S_2$, *the probability of* not *rejecting satisfies* $\mathbb{P}_{H_1}(\mathcal{R}^c) \leq p_1^{n_1} p_2^{n_2}$.

*Proof.* See Appendix B.7. □

**Experimental guidelines** In typical zero-temperature LLM evaluations, the effective support of the reference model is small (often $k \in \{2, 3\}$), while the candidate model collapses to a single dominant token, i.e., $|S_1| = k$ and $|S_2| = 1 \subset S_1$ under $H_1$.

In this practical alternative, Theorem 4.2 gives $\mathbb{P}_{H_1}(\mathcal{R}^c) \leq k^{-n_1}$, so the power depends only on the number of samples drawn from the reference model.

Under $H_0$ with $|S_1| = |S_2| = k$, false rejections occur only if at least one empirical support fails to observe all $k$ tokens. By Theorem 4.1, $\mathbb{P}_{H_0}(\mathcal{R}) \leq ke^{-n_1/k} + ke^{-n_2/k}$. The parameter $k$ therefore plays a central role, governing the tradeoff between Type-I and Type-II errors.

## 4.4. B3IT is Optimal for $k = 2$

The support-mismatch test based on $\mathcal{R}$ is intentionally conservative: it rejects only when a token observed under one model is absent from the other empirical support. This choice yields simple nonasymptotic guarantees, but can be suboptimal when the effective support size $k$ is large.

This limitation is immaterial in the zero-temperature regime we consider. Empirically, the effective support is typically very small (often $k \in \{2, 3\}$), and the dominant failure mode corresponds to a collapse of the candidate support to a singleton, i.e., $|S_1| = k$ while $|S_2| = 1 \subset S_1$.

**Theorem 4.3** (Lower bound). $H_0 : S_1 = S_2, |S_1| = 2$ vs $H_1 : |S_1| = 2, |S_2| = 1, S_2 \subset S_1$ with $n_1 = n_2 = n$. For any rejection region $\mathcal{R}$, $\mathbb{P}_{H_0}(\mathcal{R}) + \mathbb{P}_{H_1}(\mathcal{R}^c) \geq 2^{-(n+1)}$.

*Proof.* See Appendix B.8. $\square$

This lower bound formalizes the unavoidable trade-off induced by sampling noise: no test can simultaneously achieve vanishing Type-I and Type-II errors, so reducing one necessarily increases the other.

Although its assumptions may appear restrictive, they capture the dominant regime observed in our experiments, where two top logits collapse to a single mode after the model $\theta_1$ has been tampered with. Moreover, when combined with Theorem 4.1 and Theorem 4.2, this lower bound shows that the test introduced in this subsection is optimal up to a constant factor in this practical setting.

**Takeaway.** We present B3IT, a change detector that leverages BIs to implement a request-optimal change detection test. B3IT is very simple: the initialization stage identifies one (or a handful of) BIs and samples their initial distribution. The detection step samples BIs again and compares the result with the initial distribution. Thanks to the low temperature, this test simply consists in comparing the support of uniform distributions. B3IT's pseudo-code is provided in Algorithms 1 and 2.

## 5. Experimental Evaluation of B3IT

We validate B3IT both *in vitro* on a controlled benchmark, and *in vivo* on commercial LLM APIs. Our experiments address two questions: (1) can BIs be found efficiently in practice, and (2) how does B3IT compare to existing methods in accuracy and cost?

**On the existence of BIs.** In theory, logits lie in $\mathbb{R}$, so exact ties have probability zero. In practice, however, we observe many such ties. We attribute this to two factors: limited floating-point precision (causing rounding), and inference-time non-determinism (Chauvin et al., 2026; He & Thinking Machines Lab, 2025).

### 5.1. In Vitro Evaluation: TinyChange Benchmark

5.1.1. EXPERIMENTAL SETUP

We use the TinyChange benchmark (Chauvin et al., 2026), which generates controlled perturbations of open-weight LLMs, enabling systematic evaluation across varying change magnitudes.

**Models.** We evaluate on 9 instruction-tuned LLMs ranging from 0.5B to 9B parameters: Qwen2.5-0.5B, Gemma-3-

1B, Phi-4-mini, Qwen2.5-7B, DeepSeek-R1-Distill-Qwen-7B, Mistral-7B, OLMo-2-1124-7B, Llama-3.1-8B, and Gemma-2-9B. The exact model IDs can be found in Appendix E.1.

**Perturbation types.** For each model, we generate a number of *variants*, created from the TinyChange difficulty scales with these parameters: (i) **Fine-tuning**: regular or LoRA (Hu et al., 2022), for 1 epoch with 1 to 4096 single-sample steps; (ii) **Unstructured pruning**: by magnitude or random selection, removing fractions from $2^{-20}$ to 1; (iii) **Parameter noise**: Gaussian perturbations with standard deviation $\sigma \in [2^{-30}, 1]$.

**Baselines.** We compare against two black-box methods, MMLU-ALG (Chauvin et al., 2026) and MET (Gao et al., 2025), and one grey-box method, LT (Chauvin et al., 2026), which requires access to log-probabilities.

**Metrics.** A change detection method should have a good classifier performance, and a low cost. For each method, we therefore report (i) ROC AUC[1] averaged across all models and variants, and (ii) yearly cost of hourly monitoring at GPT-4.1 pricing.

**Test statistic.** The theory in Section 3 yields a binary support-mismatch test. To construct ROC curves, we need a threshold to vary. We therefore compute the total variation (TV) distance between reference and detection samples as a continuous statistic. For better performance, we also get samples from several prompts, averaging the per-prompt TV distance to obtain a test statistic.

5.1.2. INITIALIZATION STAGE

The goal of this stage is to identify BIs at minimal cost. We generate candidate prompts with minimal input length using each model's tokenizer (see Appendix E.5), then sample each candidate $m = 3$ times at $T = 0$. We use synthetic noise similar to that of LT (Chauvin et al., 2026), by mixing our queries with simulated request traffic. A prompt is selected as a BI when it yields more than one unique output across the samples.

For comprehensive evaluation, we sample 20,000 prompts per model, collecting 1,000 reference samples per identified BI (though only 50 are used per test). We obtained 105 to 1,182 BIs depending on the model.

---

[1]Area under the ROC curve, which is the plot of the true positive rate (TPR) against the false positive rate (FPR) of a binary classifier, at each threshold setting.

### 5.1.3. EXPERIMENTAL RESULTS

**Hyperparameter selection.** Figure 1 shows performance versus cost for various choices of prompt count and samples per prompt. Notably, increasing the number of prompts used can increase performance, holding the cost fixed: for instance, sampling 5 prompts 10 times outperforms sampling a single prompt 50 times. Several configurations achieve strong results; we select 5 prompts with 3 samples each, balancing accuracy and cost.

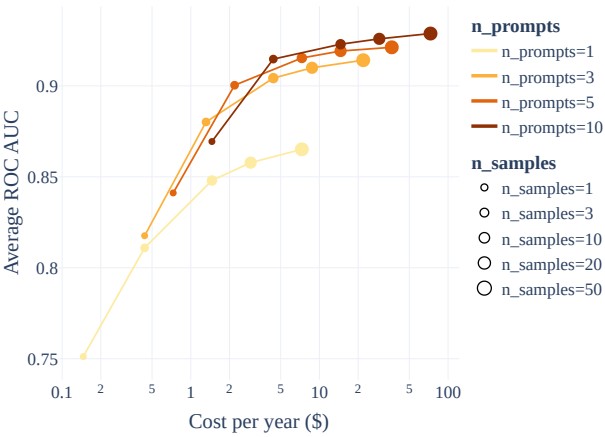

*Figure 1.* Performance increases with cost. We select 5 prompts and 3 samples/prompt for subsequent experiments.

**Comparison with baselines.** Figure 2 presents performance versus cost. To ensure fair comparison, we sweep hyperparameters for baselines (including creating $T = 0$ variants of MMLU-ALG and MET) and fix 50 reference samples per prompt, varying only detection-time parameters.

B3IT substantially outperforms black-box methods, and approaches the grey-box LT baseline despite no access to log-probabilities. At operating point ($2.2/year), B3IT achieves a ROC AUC of 0.9, whereas the next-best black-box method (MET at $T = 0$) reaches only 0.61. No black-box baseline reaches ROC AUC 0.9; the closest is MET at $T = 0$, which requires $67/year to reach 0.88.

**Performance by change difficulty.** Figure 3 shows detection performance on the fine-tuning difficulty scale. MMLU-ALG, and to a lesser extent MET, exhibit a low detection accuracy. B3IT and LT perform very well in detecting finetuned instances. As the number of fine-tuning steps decreases, the impact on model weights diminishes, increasing the detection difficulty. B3IT retains a 0.87 ROC AUC for detecting the extreme single-step fine-tuning. Refer to Appendix E.9 for the performances facing other perturbations.

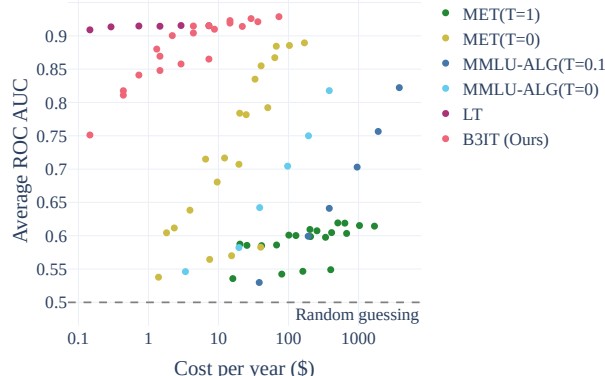

*Figure 2.* B3IT outperforms all black-box methods by a wide margin, and approaches the performance of the grey-box LT method.

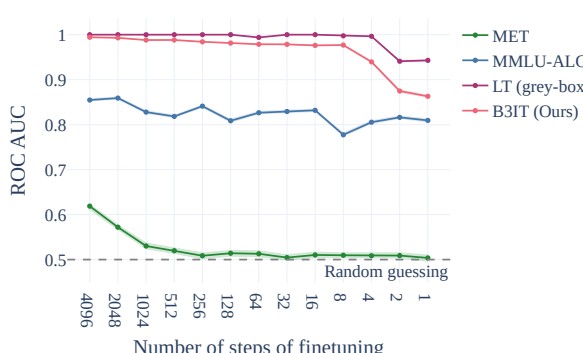

*Figure 3.* Detection performance on the TinyChange fine-tuning difficulty scale.

## 5.2. In Vivo Evaluation: Commercial LLM APIs

### 5.2.1. ENDPOINT SELECTION

Starting from all endpoints listed on OpenRouter, we exclude (i) free endpoints, which are often unreliable, and (ii) endpoints where a single-letter input produces more than 10 input tokens or more than 1 output token. From the remaining endpoints, we keep the union of the two cheapest per model and the two cheapest per provider, then drop any endpoint whose average cost ((input + output)/2) exceeds $0.5 per million tokens. This yields 131 endpoints covering 78 models across 34 providers.

### 5.2.2. PREVALENCE OF BORDER INPUTS

We searched for BIs on 131 endpoints at $T = 0$, sampling each single-token prompt 3 times. For each endpoint, we discover a working query strategy by trying, in order: (i) a plain query, (ii) a query with reasoning disabled (`reasoning.effort=none`), and (iii) queries with a

reasoning budget (`reasoning.max_tokens`) at successive powers of 2 $(1, 2, 4, \ldots, 2,048)$; the first probe to return a non-empty output defines the strategy used for that endpoint[2]. Counting only BIs found via the cheap strategies (i) and (ii), coverage rises from 73% (1k prompts) to 76% (2k prompts) of endpoints with at least one BI; also counting BIs found via strategy (iii) — the only strategy that works for these endpoints, at the cost of extra reasoning tokens per query — brings coverage to 78%. Appendix E.2 gives the full breakdown. Figure 4 reports the empirical CDF of requests required to find BIs at various temperatures, on an earlier 93-endpoint candidate set with up to 5,000 single-token prompts per endpoint, sampled 3 times each; raising the temperature beyond $T = 0$ further increases coverage, at the cost of reduced BI quality (higher prevalence implies less selective detection).

For two endpoints (`gpt-oss-20b` and `gpt-oss-120b` on amazon-bedrock), strategy discovery fails: the reasoning trace is hidden from the API response, so we cannot see the first token of output. This constitutes a limitation of methods seeking to cheaply detect changes such as B3IT.

We also study whether BIs generalize across endpoints. A Cochran-Mantel-Haenszel analysis (Appendix D) shows that BIs are strongly correlated across endpoints serving the same model from different providers (odds ratio 2.40), but essentially uncorrelated across endpoints serving different models (odds ratio 1.07). This matches the design of B3IT: BIs are sensitive to small parameter changes, and are therefore not expected to generalize across distinct models.

Finally, we observe an interesting behavior where many endpoints seem to behave differently at $T = 0$ than their limit as $T \to 0$ would indicate, which we hypothesize is due to hard-coded patterns at $T = 0$ (Anthropic, 2025).

### 5.2.3. CONTINUOUS MONITORING RESULTS

We keep the parameters of the *in vitro* experiments: 5 prompts, 50 reference samples and 3 detection samples. Continuous monitoring was set up before the prevalence study above and ran on a separate, earlier 93-endpoint candidate set. On each candidate, we searched for 5 BIs at $T = 0$ (sampling each $m = 3$ times) and selected 53 endpoints with at least one BI for monitoring[3]. We sampled them 50

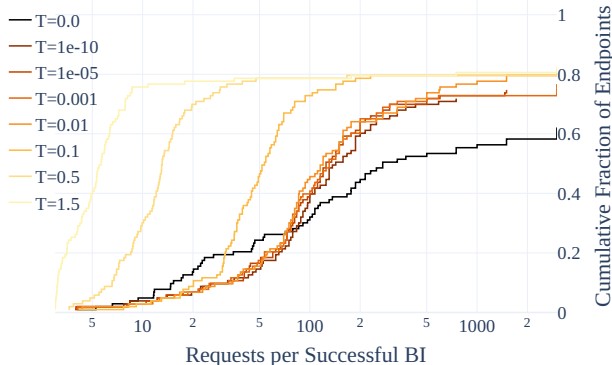

*Figure 4.* Empirical CDF of requests per BI across the earlier 93-endpoint candidate set for continuous monitoring, at various temperatures.

times to generate the reference distribution, then queried 3 detection samples every 24h over a period of 23 days.

At an average (input, output) endpoint cost of ($0.38, $1.2) / million tokens, the average cost of *hourly* monitoring would be $0.52 per endpoint per year.[4] At $0.0045 per endpoint, the cost of the initialization stage is negligible compared to the ongoing detection cost.

We identified persistent changes where the mean TV remained below a threshold of $0.5$ for at least 4 days, before crossing above for an equal period. Indeed, as suggested by our theoretical analysis, the token distribution on BIs quickly collapses to a single dominant mode under small perturbations; a TV$= 0.5$ marks the transition from a bimodal to a unimodal distribution. 8 endpoints were identified in this way (depicted on Figure 5). One of these is directly corroborated by Together AI's public changelog, which on January 29 redirected Mistral-7B-Instruct-v0.3 to the entirely different Ministral-3-14B-Instruct-2512 (Together AI, 2026); the two DeepSeek-V3-0324 changes on Hyperbolic and Atlas Cloud may also relate to a January 23 Together changelog entry redirecting that model to DeepSeek-V3.1, suggesting a similar swap may have occurred across providers.

## 6. Related Work

**Change detection of models behind APIs.** Tracking shifts in (non-LLM) ML APIs was introduced by (Chen et al., 2022), who found significant shifts in a third of the APIs they tested. For LLMs specifically, black-box methods include MET (Gao et al., 2025) (one of our baselines),

---

[2] Precisely, our BI signal is the first whitespace-delimited word of the API response, reading the reasoning trace before the visible content when both are present. For non-reasoning endpoints (strategies (i) and (ii)) this coincides with the first output token in the usual sense; for reasoning endpoints (strategy (iii)) it is the first word of the chain-of-thought. For methodological consistency and simplicity, we only look at the first word of the reasoning trace, but since the full reasoning trace is already returned, divergence could be detected at any position at no additional query cost.

[3] Endpoints with at least 1 but fewer than 5 BIs are still monitored and analyzed using all available BIs. This concerned one

endpoint with a single BI.

[4] We occasionally hit OpenRouter rate limits (HTTP 429) when aggregated client traffic is high; we retry with jitter and exponential backoff, which does not affect cost since 429 responses are not billed.

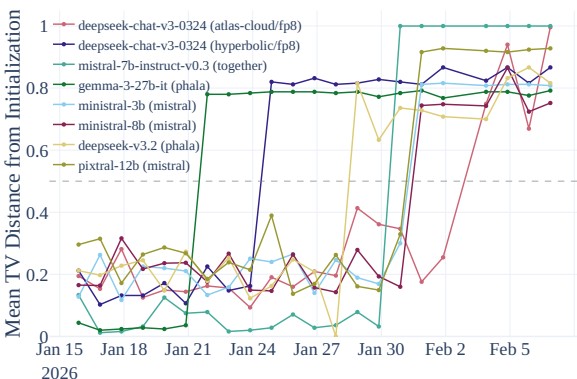

*Figure 5.* Detected changes on endpoints over 23 days.

which compared output distributions using a Maximum Mean Discrepancy test with a Hamming kernel. Daily-Bench (Phillips, 2025) attempted continuous benchmark-based monitoring of 5 LLM APIs using HELMLite (Liang et al., 2023), but was discontinued after 40 days due to substantial cost. On the white-box side (requiring access to reference model weights), ESF (Bai et al., 2025) selects sensitive inputs by analyzing penultimate-layer embeddings; TRAP (Gubri et al., 2024) optimizes adversarial suffixes via gradient-based search on a reference model; Token-DiFR (Karvonen et al., 2025) compares output tokens against a trusted reference conditioned on the same random seed, or at temperature 0. LT (Chauvin et al., 2026) operates in a grey-box setting, exploiting log probabilities for highly cost-effective monitoring. (Cai et al., 2025) re-implemented and compared several detection techniques.

**Non-determinism in LLM inference.** As change detection methods become increasingly sensitive, calibrating against hardware-induced noise becomes critical. (He & Thinking Machines Lab, 2025) showed that run-to-run LLM non-determinism stems from varying batch sizes under server load, and proposed batch-invariant kernels as a fix. (Zhang et al., 2025) leveraged within-platform determinism and cross-platform non-determinism to fingerprint the hardware and software configuration of a given model, using a logprob-based method.

**LLM fingerprinting.** LLM fingerprinting is related to change detection, but aims to identify which model is being served, typically within a closed set, and methods often attempt to remain robust to small model variations. (Pasquini et al., 2025) introduced handcrafted queries paired with a classifier to match an LLM to a known model, explicitly seeking invariance to minor changes. (Sun et al., 2025) fine-tuned embedding models on LLM-generated texts to distinguish between a set of 5 models. (Chen et al., 2024) compared explicitly different ChatGPT versions across benchmarks, which can also be seen as coarse-grained fingerprint-

ing. Change detection, by contrast, aims to trigger on *any* change, however small.

# 7. Conclusion

Change detection in LLMs behind APIs is becoming an important primitive for ensuring the reliability of downstream workflows. In this work, we established theoretical foundations for black-box change detection with a single output token, revealing a phase transition at low temperature: Border Inputs (BIs), where multiple tokens are tied at the maximal logit, yield diverging signal-to-noise ratios, making even subtle parameter changes detectable. Building on this insight, we proposed B3IT, a practical method that discovers BIs through black-box sampling and monitors them at minimal cost. Our experiments validate B3IT on both controlled benchmarks and 131 commercial endpoints across 34 providers, demonstrating detection performance comparable to grey-box methods at 1/30th the cost of the best black-box alternatives. Future work could extend the theoretical framework beyond single-token observations to sequential multi-token settings, which might unlock even more sensitive and cost-efficient detection schemes.

# 8. Acknowledgments

We would like to thank Gersende Fort, Jean-Christophe Mourrat and Matthieu Jonckheere for their help at an early stage of this project.

Timothée Chauvin, Erwan Le Merrer, François Taïani and Gilles Trédan acknowledge the support of the French Agence Nationale de la Recherche (ANR), under grant ANR-24-CE23-7787 (project PACMAM).

This project was provided with computing (AI) and storage resources by GENCI at IDRIS thanks to the grant 2025-AD011016369 on the supercomputer Jean Zay's H100 partition.

This work was supported by the Cluster SequoIA Chair FANG funded by ANR, reference number ANR-23-IACL-0009.

This work was supported by the Cluster ANITI Chair TRIAL funded by ANR, reference number n°ANR-19-P3IA-0004 and n°ANR-23-IACL-0002 and ANR Regulia ANR-23-CE23-0029.

# Impact Statement

This paper presents work whose goal is to advance the field of Machine Learning. There are many potential societal consequences of our work, none which we feel must be specifically highlighted here.

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

## A. Notations

$\Rightarrow$ denotes convergence in distribution, and more broadly the weak convergence when applied to measures of probability. For two sequences of random vectors $(U_n)$ and $(V_n)$, we write $U_n = o_{\mathbb{P}}(V_n)$ (resp. $U_n = O_{\mathbb{P}}(V_n)$) if there exists a sequence of random vectors $(W_n)$ such that $U_n = W_n V_n$ and $(W_n)$ converges in $\mathbb{P}$-probability to 0 (resp. is uniformly tight). If the base probability measure depends on $n$, we say that $(W_n)$ converges in $\mathbb{P}_n$-probability to 0 if $\forall \epsilon > 0$, $\mathbb{P}_n(\|W_n\| > \epsilon) \to 0$. Deterministic Landau notations have their usual meaning. Unless stated otherwise, $\| \cdot \|$ denotes the Euclidean norm, and for a matrix $M$, $\|M\|_{\mathrm{op}}$ and $\|M\|_{\mathrm{F}}$ denote its operator and Frobenius norms. For a symmetric matrix $A \succeq 0$, we write $\|x\|_A^2 := x^T A x$. We use $\mathrm{tr}(\cdot)$ and $\det(\cdot)$ for trace and determinant. The indicator function is denoted by $\mathbf{1}\{\cdot\}$, and $I_d$ denotes the $d \times d$ identity matrix. The Kronecker product is denoted by $\otimes$, and $\mathrm{vec}(\cdot)$ denotes vectorization (column-stacking). We write $\mathcal{N}(\mu, \Sigma)$ for the Gaussian distribution with mean $\mu$ and covariance $\Sigma$. For probability measures $P$ and $Q$ on a common measurable space, $\mathrm{TV}(P, Q)$ denotes their total variation distance. The natural logarithm is denoted by $\log$. $i$ may refer to the canonical complex number that solves "$i = \sqrt{-1}$" which should be clearly identifiable in context. $\mathbb{S}^{q-1}$ is the unit sphere in $\mathbb{R}^q$. For a set $S$, $\mathrm{Unif}(S)$ denotes the uniform distribution on $S$. The rest of the notations are either standard or introduced within text.

## B. Proofs

### B.1. Log-likelihood ratio in the LAN regime

We note $Z_n := \sqrt{n}\,(T_n - \mathbf{p}_0)$ and $\mathbf{d} := Jh$ .

We note $\mathbf{p}_0 = (p_{0,1}, \ldots, p_{0,d-1})$ and $p_{0,d} := 1 - \sum_{j=1}^{d-1} p_{0,j} > 0$. Likewise, we note $\mathbf{p}_n = (p_{n,1}, \ldots, p_{n,d-1})$ and $p_{n,d} := 1 - \sum_{j=1}^{d-1} p_{n,j} > 0$.

We write the multinomial counts $N_j := \sum_{i=1}^{n} \mathbf{1}\{Y_i = j\}$ for $j = 1, \ldots, d$, $\hat{p}_j := N_j/n$ for $j \leq d - 1$ and $\hat{p}_d := 1 - \sum_{j=1}^{d-1} \hat{p}_j = N_d/n$.

The log-likelihood ratio is ($\mathbf{p}_n$ numerator and $\mathbf{p}_0$ denominator)

$$
\begin{aligned}
\log \Lambda_n &= \sum_{j=1}^{d} N_j \log\left(\frac{p_{n,j}}{p_{0,j}}\right) \\
&= n \sum_{j=1}^{d} \hat{p}_j \log\left(\frac{p_{n,j}}{p_{0,j}}\right).
\end{aligned}
\tag{3}
$$

We define $\Delta_n := \mathbf{p}_n - \mathbf{p}_0 \in \mathbb{R}^{d-1}$ and $\Delta_{n,d} := p_{n,d} - p_{0,d} = -\mathbf{1}^T \Delta_n$. By assumption,

$$
\Delta_n = \frac{s}{\sqrt{n}} \mathbf{d} + r_n,
\tag{4}
$$

where $\sqrt{n}\|r_n\| \to 0$.

Let us start with the following Taylor expansions : For $j \leq d - 1$,

$$
\log\left(\frac{p_{n,j}}{p_{0,j}}\right) = \frac{\Delta_{n,j}}{p_{0,j}} - \frac{1}{2}\frac{\Delta_{n,j}^2}{p_{0,j}^2} + R_{n,j}, \quad R_{n,j} = O\left(\frac{|\Delta_{n,j}|^3}{p_{0,j}^3}\right),
\tag{5}
$$

and similarly for the $d$-th coordinate,

$$
\log\left(\frac{p_{n,d}}{p_{0,d}}\right) = \frac{\Delta_{n,d}}{p_{0,d}} - \frac{1}{2}\frac{\Delta_{n,d}^2}{p_{0,d}^2} + R_{n,d}, \quad R_{n,d} = O\left(\frac{|\Delta_{n,d}|^3}{p_{0,d}^3}\right).
\tag{6}
$$

Since $\mathbf{p}_0$ is interior, all $p_{0,j}$'s are bounded away from 0, hence $\sum_{j=1}^{d} |R_{n,j}| = O(\|\Delta_n\|^3)$. Multiplying by $n$ gives

$$
\begin{aligned}
n \sum_{j=1}^{d} \hat{p}_j R_{n,j} \; &\overset{\text{Holder}}{\leq} \; n (\max_j \hat{p}_j) \left( \sum_{j=1}^{d} |R_{n,j}| \right) \\
&\overset{\text{a.s.}}{\leq} n O(\|\Delta_n\|^3) \\
&\overset{\text{a.s.}}{\leq} O(n \cdot n^{-3/2}) \\
&\overset{\text{a.s.}}{\leq} O(n^{-1/2}) \\
&\leq o_{\mathbb{P}_{\mathbf{p}_0, \mathbf{p}_n}}(1),
\end{aligned}
\tag{7}
$$

because $\|\Delta_n\| = O(1/\sqrt{n})$ and because the $\hat{p}_j$'s are almost surely in $[0, 1]$.

Plugging the expansions into (3) yields

$$
\log \Lambda_n = n \sum_{j=1}^{d} \hat{p}_j \frac{\Delta_{n,j}}{p_{0,j}} - \frac{n}{2} \sum_{j=1}^{d} \hat{p}_j \frac{\Delta_{n,j}^2}{p_{0,j}^2} + o_{\mathbb{P}_{\mathbf{p}_0, \mathbf{p}_n}}(1).
\tag{8}
$$

**First-order term.** We decompose $\hat{p}_j = p_{0,j} + (\hat{p}_j - p_{0,j})$. It follows that

$$
n \sum_{j=1}^{d} \hat{p}_j \frac{\Delta_{n,j}}{p_{0,j}} = n \sum_{j=1}^{d} \Delta_{n,j} + n \sum_{j=1}^{d} (\hat{p}_j - p_{0,j}) \frac{\Delta_{n,j}}{p_{0,j}}.
\tag{9}
$$

The first sum is $n \sum_{j=1}^{d} \Delta_{n,j} = 0$ because both $\mathbf{p_n}$ and $\mathbf{p_0}$ sum to 1.

For the second sum, we write it in reduced coordinates: for $j \leq d-1$, $\hat{p}_j - p_{0,j} = Z_{n,j}/\sqrt{n}$, and for $j = d$, $\hat{p}_d - p_{0,d} = -(\mathbf{1}^T Z_n)/\sqrt{n}$. Also $\Delta_{n,d} = -(\mathbf{1}^T \Delta_n)$. Hence

$$
n \sum_{j=1}^{d} (\hat{p}_j - p_{0,j}) \frac{\Delta_{n,j}}{p_{0,j}} = \sqrt{n} \left( \sum_{j=1}^{d-1} Z_{n,j} \frac{\Delta_{n,j}}{p_{0,j}} + \left( \mathbf{1}^T Z_n \right) \frac{\mathbf{1}^T \Delta_n}{p_{0,d}} \right).
\tag{10}
$$

Using $\Delta_n = \frac{s}{\sqrt{n}} \mathbf{d} + r_n$ and $\sqrt{n} \|r_n\| \to 0$, this becomes

$$
n \sum_{j=1}^{d} \hat{p}_j \frac{\Delta_{n,j}}{p_{0,j}} = s \left( \sum_{j=1}^{d-1} Z_{n,j} \frac{d_j}{p_{0,j}} + \left( \mathbf{1}^T Z_n \right) \frac{\mathbf{1}^T \mathbf{d}}{p_{0,d}} \right) + o(1).
\tag{11}
$$

**Second-order term.** First, by the law of large numbers, under $H_0$ we have $\hat{p}_j \to p_{0,j}$ in probability.

Then, under $H_1$, Hoeffding's inequality (see Lemma A.4 in (Tsybakov, 2009)) ensures that

$$
\hat{p}_j - p_{n,j} \to 0
\tag{12}
$$

in $\mathbf{p}_n$ probability, and since $p_{n,j} \to p_{0,j}$ deterministically,

$$
\hat{p}_j - p_{0,j} \to 0
\tag{13}
$$

in probability.

In any case, $\hat{p}_j - p_{0,j} = o_{\mathbb{P}_{\mathbf{p}_0, \mathbf{p}_n}}(1)$.

So, since $\Delta_{n,j}^2 = O(1/n)$,

$$\frac{n}{2}\sum_{j=1}^d \hat{p}_j \frac{\Delta_{n,j}^2}{p_{0,j}^2} = \frac{n}{2}\sum_{j=1}^d p_{0,j}\frac{\Delta_{n,j}^2}{p_{0,j}^2} + o_{\mathbb{P}_{H_0,H_1}}(1)$$

$$= \frac{n}{2}\sum_{j=1}^d \frac{\Delta_{n,j}^2}{p_{0,j}} + o_{\mathbb{P}_{H_0,H_1}}(1) \tag{14}$$

$$= \frac{s^2}{2}\left(\sum_{j=1}^{d-1} \frac{d_j^2}{p_{0,j}} + \frac{(\mathbf{1}^T\mathbf{d})^2}{p_{0,d}}\right) + o_{\mathbb{P}_{\mathbf{p}_0,\mathbf{p}_n}}(1).$$

**Conclusion of the proof.** We have that

$$F^{-1}(\mathbf{p}_0) = \mathrm{diag}(\mathbf{p}_0)^{-1} + \frac{1}{p_{0,d}}\mathbf{1}\mathbf{1}^T. \tag{15}$$

Therefore

$$\mathbf{d}^T F^{-1}(\mathbf{p}_0)Z_n = \sum_{j=1}^{d-1} Z_{n,j}\frac{d_j}{p_{0,j}} + \left(\mathbf{1}^T Z_n\right)\frac{\mathbf{1}^T\mathbf{d}}{p_{0,d}}, \tag{16}$$

and

$$\mathbf{d}^T F^{-1}(\mathbf{p}_0)\mathbf{d} = \sum_{j=1}^{d-1} \frac{d_j^2}{p_{0,j}} + \frac{(\mathbf{1}^T\mathbf{d})^2}{p_{0,d}}. \tag{17}$$

Combining (8), (11), (14) gives

$$\log \Lambda_n = s\mathbf{d}^T F^{-1}(\mathbf{p}_0)Z_n - \frac{s^2}{2}\mathbf{d}^T F^{-1}(\mathbf{p}_0)\mathbf{d} + o_{\mathbb{P}_{\mathbf{p}_0,\mathbf{p}_n}}(1), \tag{18}$$

which concludes the proof.

## B.2. Proof of Theorem 3.1

By the Neyman-Pearson lemma (see theorem Theorem 3.2.1 in (Lehmann & Romano, 2005)), for a fixed $\alpha \in (0,1)$, the most powerful test of level $\alpha$ (which exists) to test $H_0 : \mathbf{p}_0$ vs $H_1 : \mathbf{p}_n$ has exactly Type-I error $\alpha$ and is of the form

$$\phi = \begin{cases} 1 \text{ if } \log\Lambda_n > a_n \\ c_n \text{ if } \log\Lambda_n = a_n \\ 0 \text{ if } \log\Lambda_n < a_n \end{cases} \tag{19}$$

where $c_n \in [0,1]$, $a_n \in \mathbb{R}$ satisfy $\mathbb{E}_{\mathbf{p}_0}(\phi) = \alpha$.

Under $\mathbf{p}_0$, we have $Z_n \Rightarrow \mathcal{N}(0, F(\mathbf{p}_0))$ by the central limit theorem, hence Equation (18), Slutsky's lemma and the continuous mapping theorem (see (van der Vaart, 1998)) give

$$\log \Lambda_n \Rightarrow \mathcal{N}\left(-\frac{s^2}{2}h^T J^T F(p_0)^{-1}Jh, s^2 h^T J^T F(p_0)^{-1}Jh\right). \tag{20}$$

We may expand $\mathbb{E}_{\mathbf{p}_0}(\phi)$ as

$$\mathbb{E}_{\mathbf{p}_0}(\phi) = c_n\mathbb{P}_{\mathbf{p}_0}(\log\Lambda_n = a_n) + \mathbb{P}_{\mathbf{p}_0}(\log\Lambda_n > a_n). \tag{21}$$

Let us first prove that $(a_n)$ converges to a finite limit. For the sake of contradiction, let us assume that $(a_n)$ is unbounded. We extract a subsequence $(b_n)$ that diverges (here we suppose that $b_n \to \infty$, the other case being treated in the same fashion). Without loss of generality, $\forall n, b_n > 0$. Then for every, $K > 0$, there exists a $N$ such that for every $n \geq N$, $b_n > K$. Thus for every $n \geq N$,

$$\mathbb{E}_{\mathbf{p}_0}(\phi) \leq \mathbb{P}_{\mathbf{p}_0}(\log \Lambda_n \geq K), \tag{22}$$

and we know by the portmanteau lemma (2.2 in (van der Vaart, 1998)) that

$$\mathbb{P}_{\mathbf{p}_0}(\log \Lambda_n \geq K) \to \mathbb{P}\left( N\left( -\frac{s^2}{2} h^T J^T F(\mathbf{p}_0)^{-1} Jh, s^2 h^T J^T F(\mathbf{p}_0)^{-1} Jh \right) \geq K \right), \tag{23}$$

with the right hand side that tends to $0$ when $K$ tends to infinity. This contradicts the fact that all the tests are Type-I error $\alpha$, and thus $(a_n)$ is bounded.

We now show that $(a_n)$ has a unique adherence point. Let $a_\infty$ be an adherence point of $(a_n)$. Up to an extraction, we have $a_n \to a_\infty$.

We then have, by the portmanteau lemma

$$\mathbb{P}_{\mathbf{p}_0}(\log \Lambda_n > a_n) \to \mathbb{P}\left( \mathcal{N}\left( -\frac{s^2}{2} h^T J^T F(\mathbf{p}_0)^{-1} Jh, s^2 h^T J^T F(\mathbf{p}_0)^{-1} Jh \right) > a_\infty \right). \tag{24}$$

Let $(\alpha, \beta)$ be an open interval containing $a_\infty$. Since $a_n \to a_\infty$, $a_n \in (\alpha, \beta)$ for $n$ big enough. Thus for $n$ big enough,

$$\mathbb{P}_{\mathbf{p}_0}(\log \Lambda_n = a_n) \leq \mathbb{P}_{\mathbf{p}_0}(\log \Lambda_n \in (\alpha, \beta)) \to \mathbb{P}\left( \mathcal{N}\left( -\frac{s^2}{2} h^T J^T F(\mathbf{p}_0)^{-1} Jh, s^2 h^T J^T F(\mathbf{p}_0)^{-1} Jh \right) \in (\alpha, \beta) \right) \tag{25}$$

by the portmanteau lemma. Since this holds for any neighborhood $(\alpha, \beta)$ of $a_\infty$, $\mathbb{P}_{\mathbf{p}_0}(\log \Lambda_n = a_n) \to 0$. Since we know that all the tests have same level, there can only be one such $a_\infty$

So $(a_n)$ is bounded and has a unique adherence point, it thus converges. Let us note $a_\infty$ this limit.

The last to equations also allow to calibrate $a_\infty$. Indeed since we know that the tests are of level $\alpha$, $a_\infty = -\frac{s^2}{2} h^T J^T F(\mathbf{p}_0)^{-1} Jh + \sqrt{s^2 h^T J^T F(\mathbf{p}_0)^{-1} Jh} Q_\alpha$ where $Q_\alpha$ is the quantile of order $1 - \alpha$ of $\mathcal{N}(0, 1)$.

Let us look at the asymptotic Type-II error of this sequence of optimal tests. We may expand $\mathbb{E}_{\mathbf{p}_1}(\phi)$ as

$$\begin{aligned}
1 - \mathbb{E}_{\mathbf{p}_1}(\phi) &= 1 - c_n \mathbb{P}_{\mathbf{p}_1}(\log \Lambda_n = a_n) - \mathbb{P}_{\mathbf{p}_1}(\log \Lambda_n > a_n) \\
&= \mathbb{P}_{\mathbf{p}_1}(\log \Lambda_n \leq a_n) - c_n \mathbb{P}_{\mathbf{p}_1}(\log \Lambda_n = a_n).
\end{aligned} \tag{26}$$

Recall the exact LAN expansion from Appendix B.1,

$$\log \Lambda_n = s \mathbf{d}^T F(\mathbf{p}_0)^{-1} Z_n - \frac{s^2}{2} \mathbf{d}^T F(\mathbf{p}_0)^{-1} \mathbf{d} + o_{\mathbb{P}_{\mathbf{p}_1}}(1). \tag{27}$$

Under $H_1$ we further decompose

$$Z_n = \sqrt{n}(T_n - \mathbf{p}_n) + \sqrt{n}(\mathbf{p}_n - \mathbf{p}_0). \tag{28}$$

By assumption, $\sqrt{n}(\mathbf{p}_n - \mathbf{p}_0) = s\mathbf{d} + o(1)$ deterministically. It remains to study $\underbrace{\sqrt{n}(T_n - \mathbf{p}_n))}_{=:Z'_n}$ under $\mathbb{P}_{H_1}$.

Here we would like to apply a central limit theorem to $Z'_n$ and say that it is asymptotically Gaussian. However, the fact that the base probability measure changes with $n$ is a problem that prevents the direct use of that result. We instead study this term by studying the Fourier transform of its probability measure directly and prove a CLT-like result that allows concluding the proof.

Fix $t \in \mathbb{R}^{d-1}$ and consider the characteristic function

$$\varphi_n(t) := \mathbb{E}_{\mathbf{p}_n}\left( e^{it^T Z'_n} \right) = \left( \mathbb{E}_{\mathbf{p}_n}\left( e^{it^T X_{n,1}\sqrt{n}} \right) \right)^n. \tag{29}$$

where $X_{n,1}$ is the one-hot encoding of the first categorical variable minus $\mathbf{p}_n$. Note that in the previous equation, we used the independence to transform a sum of random variables into a product of characteristic functions.

We need to control the Taylor development of this. Let $f(u) := e^{iu}$. By Taylor's theorem with integral remainder at order 2,

$$f(u) = f(0) + f'(0)u + \frac{1}{2}f''(0)u^2 + R_3(u), \quad R_3(u) = \int_0^u \frac{(u-s)^2}{2}f^{(3)}(s)ds. \tag{30}$$

Here

$$f'(u) = ie^{iu}, \quad f''(u) = -e^{iu}, \quad f^{(3)}(u) = -ie^{iu}. \tag{31}$$

Hence the remainder is exactly

$$R_3(u) = \int_0^u \frac{(u-s)^2}{2}(-i)e^{is}\,ds = -\frac{i}{2}\int_0^u (u-s)^2 e^{is}\,ds. \tag{32}$$

$$e^{iu} = 1 + iu - \frac{u^2}{2} - \frac{i}{2}\int_0^u (u-s)^2 e^{is}ds. \tag{33}$$

Thus,

$$|R_3(u)| \leq \frac{1}{2}\int_0^{|u|}(|u|-s)^2 ds = \frac{1}{2}\frac{|u|^3}{3} = \frac{|u|^3}{6}. \tag{34}$$

Applying it with $u = t^T X_{n,1}/\sqrt{n}$ yields

$$\mathbb{E}_{\mathbf{p}_n}\left(e^{it^T X_{n,1}/\sqrt{n}}\right) = 1 - \frac{1}{2n}t^T \mathbb{E}_{\mathbf{p}_n}(X_{n,1}X_{n,1}^T)t + R_n(t), \tag{35}$$

where the linear term vanishes because $\mathbb{E}_{\mathbf{p}_n}(X_{n,1}) = 0$, and where

$$|R_n(t)| \leq \mathbb{E}_{\mathbf{p}_n}\left(\left|e^{it^T X_{n,1}/\sqrt{n}} - 1 - \frac{i}{\sqrt{n}}t^T X_{n,1} + \frac{1}{2n}(t^T X_{n,1})^2\right|\right) \lesssim \frac{\|t\|^3}{n^{3/2}}, \tag{36}$$

where we used the Taylor remainder bound for $e^{ix}$ and the Hölder bound $|t^T X_{n,1}| \leq \|t\|_1$.

Hence

$$\log \varphi_n(t) = n\log\left(1 - \frac{1}{2n}t^T F(\mathbf{p}_n)t + R_n(t)\right) = -\frac{1}{2}t^T F(\mathbf{p}_n)t + o(1), \tag{37}$$

because $nR_n(t) = O(n^{-1/2}) \to 0$ for fixed $t$. Since $\mathbf{p}_n \to \mathbf{p}_0$ and $F(\cdot)$ is continuous on the interior, $F(\mathbf{p}_n) \to F(\mathbf{p}_0)$, so

$$\log \varphi_n(t) \to -\frac{1}{2}t^T F(\mathbf{p}_0)t, \tag{38}$$

which gives

$$\varphi_n(t) \to \exp\left(-\tfrac{1}{2}t^T F(\mathbf{p}_0)t\right). \tag{39}$$

Hence, if we note $\mu'_n := Z'_n \# \mathbf{p}_n$ the pushforward of $\mathbf{p}_n$ by $Z'_n$, Lévy's continuity theorem for measures (see (Ouvrard, 2009) theorem 14.8) ensures than $\mu'_n$ weakly converges to the centered Gaussian probability measure with covariance matrix $F(\mathbf{p}_0)$.

Since

$$\log \Lambda_n = s\mathbf{d}^T F(\mathbf{p}_0)^{-1}Z_n - \frac{s^2}{2}\mathbf{d}^T F(\mathbf{p}_0)^{-1}\mathbf{d} + o_{\mathbb{P}_{\mathbf{p}_n}}(1), \tag{40}$$

we can write for any fixed $t \in \mathbb{R}$

$$
\begin{aligned}
\mathbb{E}_{\mathbf{p}_n}\left(e^{it\log\Lambda_n}\right) &= \mathbb{E}_{\mathbf{p}_n}\left(e^{it\left(s\mathbf{d}^T F(\mathbf{p}_0)^{-1}Z_n - \frac{s^2}{2}\mathbf{d}^T F(\mathbf{p}_0)^{-1}\mathbf{d} + o_{\mathbb{P}_{\mathbf{p}_n}}(1)\right)}\right) \\
&= \mathbb{E}_{\mathbf{p}_n}\left(e^{it\left(s\mathbf{d}^T F(\mathbf{p}_0)^{-1}Z_n - \frac{s^2}{2}\mathbf{d}^T F(\mathbf{p}_0)^{-1}\mathbf{d} + o_{\mathbb{P}_{\mathbf{p}_n}}(1)\right)}\right) \\
&= \mathbb{E}_{\mathbf{p}_n}\left(e^{it\left(s\mathbf{d}^T F(\mathbf{p}_0)^{-1}(Z'_n + s\mathbf{d} + o(1)) - \frac{s^2}{2}\mathbf{d}^T F(\mathbf{p}_0)^{-1}\mathbf{d} + o_{\mathbb{P}_{\mathbf{p}_n}}(1)\right)}\right) \\
&= \mathbb{E}_{\mathbf{p}_n}\left(e^{it\left(s\mathbf{d}^T F(\mathbf{p}_0)^{-1}(Z'_n + s\mathbf{d}) - \frac{s^2}{2}\mathbf{d}^T F(\mathbf{p}_0)^{-1}\mathbf{d} + o_{\mathbb{P}_{\mathbf{p}_n}}(1)\right)}\right)
\end{aligned}
\tag{41}
$$

where the deterministic $o(1)$ was included in the $o$ in probability.

Let us fix $\delta > 0$ and denote by $R$ the random variable referred to as $o_{\mathbb{P}_{\mathbf{p}_1}}(1)$,

$$
\begin{aligned}
\mathbb{E}_{\mathbf{p}_n}\left(e^{it\log\Lambda_n}\right) &= \mathbb{E}_{\mathbf{p}_n}\left(e^{it\left(s\mathbf{d}^T F(\mathbf{p}_0)^{-1}(Z'_n+sd)-\frac{s^2}{2}\mathbf{d}^T F(\mathbf{p}_0)^{-1}\mathbf{d}+R\right)}\left(\mathbf{1}\{|R|>\delta\}+\mathbf{1}\{|R|\le\delta\}\right)\right) \\
&= \mathbb{E}_{\mathbf{p}_n}\left(e^{it\left(s\mathbf{d}^T F(\mathbf{p}_0)^{-1}(Z'_n+sd)-\frac{s^2}{2}\mathbf{d}^T F(\mathbf{p}_0)^{-1}\mathbf{d}+R\right)}\mathbf{1}\{|R|>\delta\}\right) \\
&\quad + \mathbb{E}_{\mathbf{p}_n}\left(e^{it\left(s\mathbf{d}^T F(\mathbf{p}_0)^{-1}(Z'_n+sd)-\frac{s^2}{2}\mathbf{d}^T F(\mathbf{p}_0)^{-1}\mathbf{d}+R\right)}\mathbf{1}\{|R|\le\delta\}\right)
\end{aligned}
\tag{42}
$$

Furthermore,

$$
\begin{aligned}
&\left|\mathbb{E}_{\mathbf{p}_n}\left(e^{it\left(s\mathbf{d}^T F(\mathbf{p}_0)^{-1}(Z'_n+sd)-\frac{s^2}{2}\mathbf{d}^T F(\mathbf{p}_0)^{-1}\mathbf{d}+R\right)}\mathbf{1}\{|R|>\delta\}\right)\right| \\
&\le \mathbb{E}_{\mathbf{p}_n}\left(\left|e^{it\left(s\mathbf{d}^T F(\mathbf{p}_0)^{-1}(Z'_n+sd)-\frac{s^2}{2}\mathbf{d}^T F(\mathbf{p}_0)^{-1}\mathbf{d}+R\right)}\right|\mathbf{1}\{|R|>\delta\}\right) \\
&\le \mathbb{E}_{\mathbf{p}_n}\left(\mathbf{1}\{|R|>\delta\}\right) \\
&\to 0
\end{aligned}
\tag{43}
$$

since $R = o_{\mathbb{P}_{\mathbf{p}_1}}(1)$.

Furthermore,

$$
\mathbb{E}_{\mathbf{p}_n}\left(\underbrace{e^{it\left(s\mathbf{d}^T F(\mathbf{p}_0)^{-1}(Z'_n+sd)-\frac{s^2}{2}\mathbf{d}^T F(\mathbf{p}_0)^{-1}\mathbf{d}+R\right)}\mathbf{1}\{|R|\le\delta\}}_{\in B(e^{it\left(s\mathbf{d}^T F(\mathbf{p}_0)^{-1}(Z'_n+sd)-\frac{s^2}{2}\mathbf{d}^T F(\mathbf{p}_0)^{-1}\mathbf{d}\right)},\delta')}\right)
\tag{44}
$$

where $B(c,r)$ is the ball in $\mathbb{C}$ centered at $c$ and of radius $r$ and where $\delta' = \sup_{-\delta\le r'\le\delta}\left|e^{it\left(s\mathbf{d}^T F(\mathbf{p}_0)^{-1}(Z'_n+sd)-\frac{s^2}{2}\mathbf{d}^T F(\mathbf{p}_0)^{-1}\mathbf{d}+r'\right)} - e^{it\left(s\mathbf{d}^T F(\mathbf{p}_0)^{-1}(Z'_n+sd)-\frac{s^2}{2}\mathbf{d}^T F(\mathbf{p}_0)^{-1}\mathbf{d}\right)}\right|$.

Thus, for any $n$ big enough, by continuity,

$$
\mathbb{E}_{\mathbf{p}_n}\left(e^{it\log\Lambda_n}\right) \in B\left(\mathbb{E}_{\mathbf{p}_n}\left(e^{it\left(s\mathbf{d}^T F(\mathbf{p}_0)^{-1}(Z'_n+sd)-\frac{s^2}{2}\mathbf{d}^T F(\mathbf{p}_0)^{-1}\mathbf{d}\right)}\right), r(\delta)\right)
\tag{45}
$$

where $r$ is non-negative and tends to 0 as $\delta$ tends to 0.

Thus, since this holds true for any $\delta$, showing the convergence of $\mathbb{E}_{\mathbf{p}_n}\left(e^{it\left(s\mathbf{d}^T F(\mathbf{p}_0)^{-1}(Z'_n+sd)-\frac{s^2}{2}\mathbf{d}^T F(\mathbf{p}_0)^{-1}\mathbf{d}\right)}\right)$ would be sufficient to show the convergence of $\mathbb{E}_{\mathbf{p}_n}\left(e^{it\log\Lambda_n}\right)$ to the same limit as it would be a bounded sequence with only one adherence point.

Furthermore, by the weak convergence of $\mu'_n$, we immediately have (see (Ouvrard, 2009) Proposition 14.5) that $\mathbb{E}_{\mathbf{p}_n}\left(e^{it\left(s\mathbf{d}^T F(\mathbf{p}_0)^{-1}(Z'_n+sd)-\frac{s^2}{2}\mathbf{d}^T F(\mathbf{p}_0)^{-1}\mathbf{d}\right)}\right)$ pointwise converges to the characteristic function of a Gaussian of mean $\frac{s^2}{2}h^T J^T F(\mathbf{p}_0)^{-1}Jh$ and of covariance matrix $s^2 h^T J^T F(\mathbf{p}_0)^{-1}Jh$.

Hence, if we note $\mu_n := \log\Lambda_n\#\mathbf{p}_n$ the pushforward of $\mathbf{p}_n$ by $\log\Lambda_n$, Lévy's continuity theorem for measures again ensures than $\mu_n$ weakly converges to a Gaussian probability measure with mean $\frac{s^2}{2}h^T J^T F(\mathbf{p}_0)^{-1}Jh$ and of covariance matrix $s^2 h^T J^T F(\mathbf{p}_0)^{-1}Jh$.

This gives that

$$
\begin{aligned}
\mathbb{P}_{\mathbf{p}_n}\left(\log\Lambda_n\le a_n\right) &= \mu_n((-\infty,a_\infty]) + \left(\mu_n((-\infty,a_n])-\mu_n((-\infty,a_\infty])\right) \\
&\to \mathbb{P}\left(\mathcal{N}\left(\frac{s^2}{2}h^T J^T F(\mathbf{p}_0)^{-1}Jh, s^2 h^T J^T F(\mathbf{p}_0)^{-1}Jh\right)\le a_\infty\right) + 0
\end{aligned}
\tag{46}
$$

where the limit of the second term comes from the fact that $a_n = a_\infty + o(1)$.

Furthermore,

$$\mathbb{P}_{\mathbf{p}_n}(\log \Lambda_n = a_n) \leq \mathbb{P}_{\mathbf{p}_n}(\log \Lambda_n \in [a_\infty - |a_n - a_\infty|, a_\infty + |a_n - a_\infty|]) \tag{47}$$
$$\to 0$$

as again $a_n = a_\infty + o(1)$.

Finally, we can compute

$$\mathbb{P}\left(\mathcal{N}\left(\frac{s^2}{2}h^T J^T F(\mathbf{p}_0)^{-1}Jh, s^2 h^T J^T F(\mathbf{p}_0)^{-1}Jh\right) \leq a_\infty\right)$$
$$= \mathbb{P}\left(\mathcal{N}\left(\frac{s^2}{2}h^T J^T F(\mathbf{p}_0)^{-1}Jh, s^2 h^T J^T F(\mathbf{p}_0)^{-1}Jh\right) \leq -\frac{s^2}{2}h^T J^T F(\mathbf{p}_0)^{-1}Jh + \sqrt{s^2 h^T J^T F(\mathbf{p}_0)^{-1}Jh}Q_\alpha\right)$$
$$= \mathbb{P}\left(\mathcal{N}(0,1) \leq -\sqrt{s^2 h^T J^T F(\mathbf{p}_0)^{-1}Jh} + Q_\alpha\right), \tag{48}$$

which concludes the proof.

### B.3. Proof of Lemma 3.2

For $p_i^{(\tau)} = \exp(z_i/\tau)/\sum_j \exp(z_j/\tau)$, we have

$$\frac{\partial p_i^{(\tau)}}{\partial z_k} = \frac{1}{\tau} p_i^{(\tau)}(\mathbf{1}\{i = k\} - p_k^{(\tau)}). \tag{49}$$

So, the full Jacobian w.r.t. the logits is the $d \times d$ matrix

$$\frac{\partial p^{(\tau)}}{\partial z} = \frac{1}{\tau}\left(\mathrm{diag}(\mathbf{p}^{(\tau)}) - \mathbf{p}^{(\tau)}(\mathbf{p}^{(\tau)})^T\right) \tag{50}$$
$$= \frac{1}{\tau}\Sigma(\mathbf{p}^{(\tau)}).$$

Since we work on reduced coordinates, we only keep the first $d - 1$ coordinates of $\mathbf{p}^{(\tau)}$. Thus the Jacobian of the reduced vector $p_{1:(d-1)}^{(\tau)}$ w.r.t. $z$ is the $(d-1) \times d$ matrix obtained by keeping the first $d - 1$ rows of $\Sigma(\mathbf{p}^{(\tau)})$:

$$\frac{\partial p_{1:(d-1)}^{(\tau)}}{\partial z} = \frac{1}{\tau}A(\mathbf{p}^{(\tau)}), \tag{51}$$

where $A(\mathbf{p})$ has entries $A_{c,k}(\mathbf{p}) = p_c(\mathbf{1}\{c = k\} - p_k)$ for $c \leq d - 1$.

By the chain rule, this gives

$$J^{(\tau)}(\theta) = \frac{1}{\tau}A\left(\mathbf{p}^{(\tau)}(\theta)\right)J_z(\theta). \tag{52}$$

Then we claim that we also have the identity

$$A(\mathbf{p})^T F(\mathbf{p}_{1:(d-1)})^{-1}A(\mathbf{p}) = \Sigma(\mathbf{p}). \tag{53}$$

Indeed, let us first observe that

$$F(\mathbf{p}_{1:(d-1)})^{-1} = \mathrm{diag}(\mathbf{p}_{1:(d-1)})^{-1} + \frac{1}{p_d}\mathbf{1}\mathbf{1}^T. \tag{54}$$

This formula can be obtained with the Sherman-Morrison formula, but we only verify its validity below.

Let $D := \mathrm{diag}(\mathbf{p}_{1:(d-1)})$ and $u := \mathbf{p}_{1:(d-1)}$, so that $F := F(\mathbf{p}_{1:(d-1)}) = D - uu^T$.

We set $M := D^{-1} + \frac{1}{p_d}\mathbf{1}\mathbf{1}^T$. We now verify that $FM = I_{d-1}$.

First,

$$DD^{-1} = I \tag{55}$$

and

$$D\left(\frac{1}{p_d}\mathbf{1}\mathbf{1}^T\right) = \frac{1}{p_d}u\,\mathbf{1}^T. \tag{56}$$

Next,

$$
\begin{aligned}
uu^T D^{-1} &= u(u^T D^{-1}) \\
&= u\mathbf{1}^T
\end{aligned}
\tag{57}
$$

and finally

$$
\begin{aligned}
uu^T\left(\frac{1}{p_d}\mathbf{1}\mathbf{1}^T\right) &= \frac{1}{p_d}u(u^T\mathbf{1})\mathbf{1}^T \\
&= \frac{1}{p_d}u(1 - p_d)\mathbf{1}^T.
\end{aligned}
\tag{58}
$$

Therefore,

$$
\begin{aligned}
FM &= (D - uu^T)\left(D^{-1} + \frac{1}{p_d}\mathbf{1}\mathbf{1}^T\right) \\
&= \underbrace{DD^{-1}}_{I} + \underbrace{D\frac{1}{p_d}\mathbf{1}\mathbf{1}^T}_{\frac{1}{p_d}u\mathbf{1}^T} - \underbrace{uu^T D^{-1}}_{u\mathbf{1}^T} - \underbrace{uu^T\frac{1}{p_d}\mathbf{1}\mathbf{1}^T}_{\frac{1}{p_d}(1-p_d)u\mathbf{1}^T} \\
&= I + \left(\frac{1}{p_d} - 1 - \frac{1 - p_d}{p_d}\right)u\mathbf{1}^T = I.
\end{aligned}
\tag{59}
$$

Thus (54) holds.

We now compute $A^T F^{-1} A$. Using $A = [F \mid -p_d p_{1:(d-1)}]$, we have

$$
\begin{aligned}
A^T F^{-1} A &= \begin{bmatrix} F^T \\ (-p_d\mathbf{p}_{1:(d-1)})^T \end{bmatrix} F^{-1}\begin{bmatrix} F \mid -p_d\mathbf{p}_{1:(d-1)} \end{bmatrix} \\
&= \begin{bmatrix} F & -p_d\mathbf{p}_{1:(d-1)} \\ -p_d\mathbf{p}_{1:(d-1)}^T & p_d^2\mathbf{p}_{1:(d-1)}^T F^{-1}\mathbf{p}_{1:(d-1)} \end{bmatrix},
\end{aligned}
\tag{60}
$$

since $F$ is symmetric.

It remains to compute $\mathbf{p}_{1:(d-1)}^T F^{-1}\mathbf{p}_{1:(d-1)}$ using (54):

$$
\begin{aligned}
\mathbf{p}_{1:(d-1)}^T F^{-1}\mathbf{p}_{1:(d-1)} &= \mathbf{p}_{1:(d-1)}^T \operatorname{diag}(\mathbf{p}_{1:(d-1)})^{-1}\mathbf{p}_{1:(d-1)} + \frac{1}{p_d}\mathbf{p}_{1:(d-1)}^T \mathbf{1}\mathbf{1}^T\mathbf{p}_{1:(d-1)} \\
&= (1 - p_d) + \frac{(1 - p_d)^2}{p_d} \\
&= \frac{1 - p_d}{p_d}.
\end{aligned}
\tag{61}
$$

Hence the bottom-right entry is

$$
\begin{aligned}
p_d^2\mathbf{p}_{1:(d-1)}^T F^{-1}\mathbf{p}_{1:(d-1)} &= p_d^2\frac{1 - p_d}{p_d} \\
&= p_d(1 - p_d) \\
&= p_d - p_d^2.
\end{aligned}
\tag{62}
$$

Therefore,

$$
A^T F^{-1} A = \begin{bmatrix} \operatorname{diag}(\mathbf{p}_{1:(d-1)}) - \mathbf{p}_{1:(d-1)}\mathbf{p}_{1:(d-1)}^T & -p_d\mathbf{p}_{1:(d-1)} \\ -p_d\mathbf{p}_{1:(d-1)}^T & p_d - p_d^2 \end{bmatrix}.
\tag{63}
$$

which is exactly $\operatorname{diag}(\mathbf{p}) - \mathbf{p}\mathbf{p}^T = \Sigma(p)$.

By definition,

$$\text{SNR}^2(h) = h^T \left( (J^{(\tau)})^T F(\mathbf{p}_{1:(d-1)}^{(\tau)})^{-1} J^{(\tau)} \right) h. \tag{64}$$

Substituting (52) gives

$$\text{SNR}^2(h) = \frac{1}{\tau^2} h^T \left( J_z^T (A^T F^{-1} A) J_z \right) h = \frac{1}{\tau^2} h^T \left( J_z^T \Sigma(\mathbf{p}^{(\tau)}) J_z \right) h, \tag{65}$$

which is (2).

## B.4. Proof of Theorem 3.3

If $k = 1$, let $\mathcal{M} = \{i^\star\}$ and $\Delta = \min_{j \neq i^\star}(z_{i^\star} - z_j) > 0$, then for $j \neq i^\star$,

$$\frac{p_j^{(\tau)}}{p_{i^\star}^{(\tau)}} = \exp\left( -\frac{z_{i^\star} - z_j}{\tau} \right) \tag{66}$$
$$\leq \exp(-\Delta/\tau),$$

so $p_{i^\star}^{(\tau)} = 1 - O(\exp(-\Delta/\tau))$ and thus $1 - \|\mathbf{p}^{(\tau)}\|_2^2 = O(\exp(-\Delta/\tau))$.

Since $\Sigma(\mathbf{p}^{(\tau)})$ is positive semidefinite and $\|\Sigma(\mathbf{p}^{(\tau)})\|_{\text{op}} \leq \operatorname{tr}(\Sigma(\mathbf{p}^{(\tau)})) = 1 - \|\mathbf{p}^{(\tau)}\|_2^2$,

$$h^T J_z^T \Sigma(\mathbf{p}^{(\tau)}) J_z h \leq \|J_z h\|^2 \|\Sigma(\mathbf{p}^{(\tau)})\|_{\text{op}} \tag{67}$$
$$= O(\exp(-\Delta/\tau)).$$

Plugging into (2) gives $\text{SNR}^2(h) \to 0$.

If $k \geq 2$, by continuity

$$h^T \left( J_z^T \Sigma(\mathbf{p}^{(\tau)}) J_z \right) h \to h^T \left( J_z^T \Sigma_{\mathcal{M}} J_z \right) h \tag{68}$$

Plugging into (2) gives $\text{SNR}^2(h) \to \infty$ if $h^T \left( J_z^T \Sigma_{\mathcal{M}} J_z \right) h \neq 0$.

## B.5. Effect of the last layer

Let

$$A := J_z(\theta)^T \Sigma_{\mathcal{M}} J_z(\theta). \tag{69}$$

Since $\Sigma_{\mathcal{M}} \succeq 0$, we have $A \succeq 0$. In particular,

$$\{h \in \mathbb{S}^{q-1} : h^T A h = 0\} = \mathbb{S}^{q-1} \cap \ker(A) \tag{70}$$

has measure zero as soon as $A \neq 0$. It is therefore sufficient to show that $A$ is not the zero matrix.

To this end, it suffices to prove that $\operatorname{tr}(A) > 0$. Recall the Jacobian decomposition

$$J_z(\theta) = \left[ W J_r(\theta_{\text{pre}}) \,\middle|\, r(\theta_{\text{pre}}, x_{\text{test}})^T \otimes I_d \,\middle|\, I_d \right]. \tag{71}$$

We define the head contribution to the Jacobian by

$$J_{z,\text{head}} := \left[ r(\theta_{\text{pre}}, x_{\text{test}})^T \otimes I_d \,\middle|\, I_d \right]. \tag{72}$$

We drop the (nonnegative) pre-head contribution, which yields

$$\operatorname{tr}(A) = \operatorname{tr}(J_z^T \Sigma_{\mathcal{M}} J_z) \tag{73}$$
$$\geq \operatorname{tr}(J_{z,\text{head}}^T \Sigma_{\mathcal{M}} J_{z,\text{head}}),$$

which gives

$$\text{tr}\big(J_{z,\text{head}}^T \Sigma_{\mathcal{M}} J_{z,\text{head}}\big) = \text{tr}\big((r^T \otimes I_d)^T \Sigma_{\mathcal{M}}(r^T \otimes I_d)\big) + \text{tr}\big(I_d^T \Sigma_{\mathcal{M}} I_d\big). \tag{74}$$

For the bias block,

$$\text{tr}\big(I_d^T \Sigma_{\mathcal{M}} I_d\big) = \text{tr}(\Sigma_{\mathcal{M}}) = 1 - \|\mathbf{p}_{\mathcal{M}}\|_2^2, \tag{75}$$

where we recall that $\mathbf{p}_{\mathcal{M}}$ assigns uniform mass to $\mathcal{M}$ and zero elsewhere.

For the weight block, by the properties of the Kronecker product,

$$
\begin{aligned}
(r^T \otimes I_d)^T \Sigma_{\mathcal{M}}(r^T \otimes I_d) &= (r^T \otimes I_d)^T \Sigma_{\mathcal{M}}(r^T \otimes I_d) \\
&= ((r^T)^T \otimes I_d^T)\Sigma_{\mathcal{M}}(r^T \otimes I_d) \\
&= (r \otimes I_d)\Sigma_{\mathcal{M}}(r^T \otimes I_d) \\
&= ((r \otimes I_d)((1) \otimes \Sigma_{\mathcal{M}}))(r^T \otimes I_d) \\
&= (r(1) \otimes I_d \Sigma_{\mathcal{M}})(r^T \otimes I_d) \\
&= (r \otimes \Sigma_{\mathcal{M}})(r^T \otimes I_d) \\
&= (rr^T) \otimes \Sigma_{\mathcal{M}},
\end{aligned}
\tag{76}
$$

where $r := r(\theta_{\text{pre}}, x_{\text{test}})$, which yields

$$
\begin{aligned}
\text{tr}\big((rr^T) \otimes \Sigma_{\mathcal{M}}\big) &= \text{tr}(rr^T)\,\text{tr}(\Sigma_{\mathcal{M}}) \\
&= \|r\|_2^2\,(1 - \|\mathbf{p}_{\mathcal{M}}\|_2^2).
\end{aligned}
\tag{77}
$$

Thus,

$$\text{tr}\big(J_{z,\text{head}}^T \Sigma_{\mathcal{M}} J_{z,\text{head}}\big) = (\|r\|_2^2 + 1)\,(1 - \|\mathbf{p}_{\mathcal{M}}\|_2^2). \tag{78}$$

Since $\mathbf{p}_{\mathcal{M}}$ is uniform on $\mathcal{M}$ of size $k$, we have $\|\mathbf{p}_{\mathcal{M}}\|_2^2 = 1/k$, and thus

$$1 - \|\mathbf{p}_{\mathcal{M}}\|_2^2 = 1 - \frac{1}{k} > 0 \tag{79}$$

for $k \geq 2$. Therefore (78) is strictly positive for $k \geq 2$, implying $\text{tr}(A) > 0$ and hence $A \neq 0$. Consequently,

$$h^T\big(J_z^T \Sigma_{\mathcal{M}} J_z\big)h \neq 0 \tag{80}$$

for almost every $h \in \mathbb{S}^{q-1}$.

In particular, if $h$ is drawn from any non-degenerate continuous distribution on $\mathbb{S}^{q-1}$ (or in $\mathbb{R}^q$ and normalized), the condition holds with probability one.

### B.6. Proof of Theorem 4.1

If both empirical supports equal the true support, i.e., if $\hat{S}_1 = S$ and $\hat{S}_2 = S$, then necessarily $\hat{S}_1 = \hat{S}_2$, hence $\mathcal{R}$ cannot occur. So,

$$\mathcal{R} \subseteq \{\hat{S}_1 \neq S\} \cup \{\hat{S}_2 \neq S\}. \tag{81}$$

By the union bound,

$$\mathbb{P}_{H_0}(\mathcal{R}) \leq \mathbb{P}(\hat{S}_1 \neq S) + \mathbb{P}(\hat{S}_2 \neq S).$$

Fix $i \in \{1, 2\}$. The event $\{\hat{S}_i \neq S\}$ means that at least one element of $S$ was never observed among the $n_i$ draws. For each token $a \in S$, we define the indicator

$$I_{a,i} := \mathbf{1}\{a \notin \hat{S}_i\}. \tag{82}$$

Then $\{\hat{S}_i \neq S\} = \{\sum_{a \in S} I_{a,i} \geq 1\}$, so by Markov's inequality,

$$
\begin{aligned}
\mathbb{P}(\hat{S}_i \neq S) &= \mathbb{P}\Big(\sum_{a \in S} I_{a,i} \geq 1\Big) \\
&\leq \mathbb{E}\Big(\sum_{a \in S} I_{a,i}\Big) \\
&= \sum_{a \in S} \mathbb{P}(a \notin \hat{S}_i).
\end{aligned}
\tag{83}
$$

Under $H_0$, each draw hits a fixed token $a \in S$ with probability $1/k$, so the probability that $a$ is never observed in $n_i$ i.i.d. draws is

$$\mathbb{P}(a \notin \hat{S}_i) = \left(1 - \frac{1}{k}\right)^{n_i}. \tag{84}$$

So,

$$\mathbb{P}(\hat{S}_i \neq S) \leq k\left(1 - \frac{1}{k}\right)^{n_i}. \tag{85}$$

Combining this for $i = 1$ and $i = 2$ gives

$$\mathbb{P}_{H_0}(\mathcal{R}) \leq k\left(1 - \frac{1}{k}\right)^{n_1} + k\left(1 - \frac{1}{k}\right)^{n_2}. \tag{86}$$

### B.7. Proof of Theorem 4.2

If the test does not reject, then all observed samples must belong to the intersection $I = S_1 \cap S_2$. In particular,

$$\mathcal{R}^c \subseteq \{\forall i, X_i \in I\} \cap \{\forall j, Y_j \in I\}. \tag{87}$$

By independence,

$$\mathbb{P}_{H_1}(\mathcal{R}^c) \leq \mathbb{P}(\forall i, X_i \in I)\,\mathbb{P}(\forall j, Y_j \in I). \tag{88}$$

Since $X_i \sim \mathrm{Unif}(S_1)$ and $Y_j \sim \mathrm{Unif}(S_2)$, $\mathbb{P}(X_i \in I) = \frac{|I|}{k_1} = p_1$ and $\mathbb{P}(Y_j \in I) = \frac{|I|}{k_2} = p_2$, which yields

$$\mathbb{P}_{H_1}(\mathcal{R}^c) \leq p_1^{n_1} p_2^{n_2}. \tag{89}$$

### B.8. Proof of Theorem 4.3

We prove the lower bound using a two-point argument based on Le Cam's method.

We fix two distinct tokens $a \neq b$ in $\Omega$ and we consider the following pair of hypotheses: $H_0 : S_1^{(0)} = S_2^{(0)} = \{a, b\}$ and $H_1 : S_1^{(1)} = \{a, b\}$, $S_2^{(1)} = \{a\}$.

Let $Z := (X_1, \ldots, X_n, Y_1, \ldots, Y_n)$ denote the full observation, where the samples are independent and satisfy $X_i \overset{\text{i.i.d.}}{\sim} \mathrm{Unif}(S_1^{(l)})$ and $Y_j \overset{\text{i.i.d.}}{\sim} \mathrm{Unif}(S_2^{(l)})$ for $l \in \{0, 1\}$.

Let $\mathbb{P}_0$ and $\mathbb{P}_1$ denote the laws of $Z$ under the null and alternative hypotheses, respectively.

Under both hypotheses, the reference support is the same, $S_1^{(0)} = S_1^{(1)} = \{a, b\}$. Consequently, the distribution of the $X$-sample $(X_1, \ldots, X_n)$ is identical under $\mathbb{P}_0$ and $\mathbb{P}_1$. Since the samples are independent, the total variation distance between $\mathbb{P}_0$ and $\mathbb{P}_1$ reduces to that between the marginal distributions of the $Y$ samples, which gives

$$\mathrm{TV}(\mathbb{P}_0, \mathbb{P}_1) = \mathrm{TV}\left((\mathrm{Unif}\{a, b\})^{\otimes n}, \delta_a^{\otimes n}\right), \tag{90}$$

where $\delta_a^{\otimes n}$ denotes the point mass on the constant sequence $(a, \ldots, a)$.

Since $\delta_a^{\otimes n}$ is supported on a single point, we have

$$\mathrm{TV}\left((\mathrm{Unif}\{a, b\})^{\otimes n}, \delta_a^{\otimes n}\right) = 1 - (\mathrm{Unif}\{a, b\})^{\otimes n}(a, \ldots, a). \tag{91}$$

Under $(\mathrm{Unif}\{a, b\})^{\otimes n}$, the probability of observing $(a, \ldots, a)$ is $2^{-n}$, hence

$$\mathrm{TV}(\mathbb{P}_0, \mathbb{P}_1) = 1 - 2^{-n}, \tag{92}$$

Le Cam's inequality (Tsybakov, 2009) states that for any rejection region $\mathcal{R}$,

$$\mathbb{P}_0(\mathcal{R}) + \mathbb{P}_1(\mathcal{R}^c) \geq \frac{1}{2}\left(1 - \mathrm{TV}(\mathbb{P}_0, \mathbb{P}_1)\right). \tag{93}$$

which gives the result.

# C. Derivation of Optimal Sampling Budget

In this section, we derive the optimal stopping limit $m$ (maximum number of samples per candidate) to minimize the total query budget required to identify $N$ Border Inputs.

## C.1. Problem Setup

Let $f_B$ be the fraction of candidate inputs that produce a Bernoulli distribution ($p = 0.5$), and let $1 - f_B$ be the fraction of inputs that produce a deterministic Dirac distribution. We define the cost function $\mathcal{L}(m)$ as the *expected number of samples required to identify a single Bernoulli candidate*. This is given by the ratio of the expected samples spent per candidate to the probability that a specific candidate is identified as a success:

$$\mathcal{L}(m) = \frac{\mathbb{E}[S_m]}{P(\text{success}|m)} \tag{94}$$

where $\mathbb{E}[S_m]$ is the expected number of samples taken per candidate given a limit $m$, and $P(\text{success}|m)$ is the probability a candidate is confirmed as Bernoulli within $m$ samples.

## C.2. Cost Function Derivation

A candidate is identified as a success if it is a Bernoulli distribution and we observe at least two differing outputs within $m$ samples. The probability of failing to identify a Bernoulli distribution (observing $m$ identical outputs) is $2 \cdot (1/2)^m = (1/2)^{m-1}$. Thus:

$$P(\text{success}|m) = f_B \left[ 1 - \left( \frac{1}{2} \right)^{m-1} \right] \tag{95}$$

The expected number of samples $\mathbb{E}[S_m]$ depends on the candidate type:

- **Dirac Candidates (prob. $1 - f_B$):** We always sample $m$ times before discarding (as outputs are identical). Cost is $m$.

- **Bernoulli Candidates (prob. $f_B$):** We stop early if variation is observed. The expected number of samples is capped at $m$. Using the geometric series for the expected stopping time, the cost is $3 - (1/2)^{m-2}$.

Combining these terms, the total expected cost per candidate is:

$$\mathbb{E}[S_m] = (1 - f_B)m + f_B \left[ 3 - \left( \frac{1}{2} \right)^{m-2} \right] \tag{96}$$

Substituting into the objective function:

$$\mathcal{L}(m) = \frac{(1 - f_B)m + f_B \left[ 3 - (1/2)^{m-2} \right]}{f_B \left[ 1 - (1/2)^{m-1} \right]} \tag{97}$$

## C.3. Comparison of Strategies

We compare the Naive Strategy ($m = 2$) against increasing the limit to $m = 3$.

**Case $m = 2$:**

$$\mathcal{L}(2) = \frac{4}{f_B} \tag{98}$$

**Case $m = 3$:**

$$\mathcal{L}(3) = \frac{4}{f_B} - \frac{2}{3} \tag{99}$$

Since $\mathcal{L}(3) < \mathcal{L}(2)$ holds for all $f_B \in (0, 1]$, the strategy $m = 3$ is strictly superior to $m = 2$.

---

**Algorithm 1** B3IT Initialization

---

**Require:** Candidate inputs $\mathcal{X} = \{x_1, \ldots, x_n\}$, samples per input $m$, target number of BIs $\bar{n}$, reference samples $n_1$, temperature $T \approx 0$
**Ensure:** Border Inputs $\mathcal{B}$, reference distributions $\{\hat{S}_1(x) : x \in \mathcal{B}\}$
 1: **// Border Input Discovery**
 2: $\mathcal{B} \leftarrow \emptyset$
 3: **for** $x \in \mathcal{X}$ **do**
 4:     Sample outputs $\{y_1, \ldots, y_m\} \leftarrow \text{Query}(x, T, m)$
 5:     **if** $|\{y_1, \ldots, y_m\}| > 1$ **then**
 6:         $\mathcal{B} \leftarrow \mathcal{B} \cup \{x\}$
 7:     **end if**
 8:     **if** $|\mathcal{B}| \geq \bar{n}$ **then**
 9:         **break**
10:     **end if**
11: **end for**
12: **// Reference Distribution Estimation**
13: **for** $x \in \mathcal{B}$ **do**
14:     Sample reference outputs $\{y_1^{(1)}, \ldots, y_{n_1}^{(1)}\} \leftarrow \text{Query}(x, T, n_1)$
15:     $\hat{S}_1(x) \leftarrow \{y_1^{(1)}, \ldots, y_{n_1}^{(1)}\}$
16: **end for**
17: **return** $\mathcal{B}, \{\hat{S}_1(x) : x \in \mathcal{B}\}$

---

**Algorithm 2** B3IT Detection

---

**Require:** Border Inputs $\mathcal{B}$, reference distributions $\{\hat{S}_1(x) : x \in \mathcal{B}\}$, detection samples $n_2$, temperature $T \approx 0$
**Ensure:** Change detection decision
 1: **for** $x \in \mathcal{B}$ **do**
 2:     Sample detection outputs $\{y_1^{(2)}, \ldots, y_{n_2}^{(2)}\} \leftarrow \text{Query}(x, T, n_2)$
 3:     $\hat{S}_2(x) \leftarrow \{y_1^{(2)}, \ldots, y_{n_2}^{(2)}\}$
 4:     **if** $\hat{S}_1(x) \triangle \hat{S}_2(x) \neq \emptyset$ **then**
 5:         **return** CHANGE DETECTED
 6:     **end if**
 7: **end for**
 8: **return** NO CHANGE DETECTED

---

### C.4. Optimal Condition for $m = 4$

We next determine when increasing the budget to $m = 4$ becomes beneficial by solving for $\mathcal{L}(4) < \mathcal{L}(3)$. For $m = 4$, the cost function becomes:

$$\mathcal{L}(4) = \frac{4 - 1.25 f_B}{0.875 f_B} \tag{100}$$

Setting the inequality $\mathcal{L}(4) < \mathcal{L}(3)$:

$$\frac{4 - 1.25 f_B}{0.875 f_B} < \frac{4}{f_B} - \frac{2}{3}$$
$$f_B > \frac{3}{4} \tag{101}$$

Thus, $m = 4$ is the optimal strategy only if the prevalence of Bernoulli candidates $f_B > 0.75$. Assuming a sparse distribution of valid candidates ($f_B \ll 0.75$), $m = 3$ is the global optimum.

## D. Generalization of Border Inputs across endpoints

A natural question is whether Border Inputs found on one endpoint are also Border Inputs on another endpoint. We test this using the *in vivo* data from the prevalence study, and apply the Cochran-Mantel-Haenszel (CMH) test on ordered pairs of endpoints $(A, B)$. The CMH odds ratio quantifies how much more likely a BI on $A$ is to also be a BI on $B$, relative to a non-BI on $A$.

We split the analysis into two regimes: pairs of endpoints serving the *same model* from different providers, and pairs serving *different models*. Table 1 reports the results.

| Pair type | # Models | # Endpoints | CMH odds ratio (95% CI) | $p$ |
|---|---|---|---|---|
| Same model, different provider | 28 | 86 | 2.40 (2.12, 2.72) | $1.4 \times 10^{-49}$ |
| Different model | 55 | 104 | 1.07 (1.04, 1.11) | $1.2 \times 10^{-4}$ |

*Table 1.* Cochran-Mantel-Haenszel test for Border Input co-occurrence across endpoint pairs

For same-model pairs, a BI on one endpoint is $2.4\times$ more likely to also be a BI on another endpoint serving the same model. This correlation is expected, since the model is supposed to be the same.

For different-model pairs, the effect is much smaller: $1.07\times$, very close to independence. The fact that BIs don't generalize across models relates to their effectiveness as change detectors: by design, they are very likely to break down on a model change, which also means they are unlikely to generalize across models.

# E. Experimental addendum

## E.1. Models used in the TinyChange Benchmark

- Qwen/Qwen2.5-0.5B-Instruct

- Qwen/Qwen2.5-7B-Instruct

- google/gemma-3-1b-it

- google/gemma-2-9b-it

- microsoft/Phi-4-mini-instruct

- mistralai/Mistral-7B-Instruct-v0.3

- deepseek-ai/DeepSeek-R1-Distill-Qwen-7B

- meta-llama/Llama-3.1-8B-Instruct

- allenai/OLMo-2-1124-7B-Instruct

## E.2. Border Input prevalence coverage

*Table 2.* Fraction of endpoints for which at least $k$ Border Inputs were found, over $N = 131$ endpoints, under three query budgets: 1k single-token prompts (excluding reasoning endpoints), 2k single-token prompts (excluding reasoning), and 2k single-token prompts including reasoning endpoints. Each prompt is sampled three times.

|  | 1k prompts | 2k prompts | 2k prompts + reasoning |
|---|---|---|---|
| $\geq 1$ BIs | 95/131 (73%) | 99/131 (76%) | 102/131 (78%) |
| $\geq 2$ BIs | 94/131 (72%) | 97/131 (74%) | 98/131 (75%) |
| $\geq 3$ BIs | 90/131 (69%) | 93/131 (71%) | 94/131 (72%) |
| $\geq 4$ BIs | 87/131 (66%) | 92/131 (70%) | 93/131 (71%) |
| $\geq 5$ BIs | 86/131 (66%) | 90/131 (69%) | 91/131 (69%) |

## E.3. Endpoints used for the Border Input prevalence study

The following $N = 131$ endpoints were used for the Border Input prevalence study. For each, we report the number of BIs found and the total number of API requests sent. Endpoints whose label includes `reasoning` are reasoning endpoints, annotated with the strategy used to obtain their first output token: `disabled` means reasoning was turned off via `reasoning.effort=none`; `budget=N` means the endpoint only produces visible output once at least $N$ reasoning tokens have been generated (requested via `reasoning.max_tokens=N`), so each query consumes extra reasoning tokens. 2 endpoints are annotated `hidden reasoning` (counted as 0 BIs in the totals above): they reasoned but did not expose the reasoning content in the API response, preventing us from sampling their first output token.

```
[BIs: 206, 3000 reqs                      ] model: z-ai/glm-4.7-flash  provider: phala
[BIs: 189, 1005 reqs                      ] model: qwen/qwen-2.5-7b-instruct  provider: atlas-cloud/fp8
[BIs: 157, 999 reqs                       ] model: deepseek/deepseek-v3.2  provider: deepinfra/fp4
[BIs: 136, 2533 reqs                      ] model: qwen/qwen3-coder-30b-a3b-instruct  provider: siliconflow/fp8
[BIs: 121, 1056 reqs                      ] model: meta-llama/llama-4-scout  provider: groq
[BIs: 119, 447 reqs                       ] model: qwen/qwen3.5-9b  provider: venice/fp8
[BIs: 107, 1445 reqs                      ] model: meituan/longcat-flash-chat  provider: atlas-cloud/fp8
[BIs: 106, 966 reqs                       ] model: deepseek/deepseek-chat-v3-0324  provider: deepinfra/fp4
[BIs: 95, 2928 reqs                       ] model: qwen/qwen3-235b-a22b-2507  provider: deepinfra/fp8
[BIs: 86, 978 reqs                        ] model: qwen-turbo  provider: alibaba
[BIs: 72, 942 reqs                        ] model: deepseek/deepseek-v3.2  provider: atlas-cloud/fp8
[BIs: 71, 1767 reqs                       ] model: qwen/qwen3-30b-a3b-instruct-2507  provider: siliconflow/fp8
[BIs: 61, 945 reqs                        ] model: meta-llama/llama-4-scout  provider: deepinfra/fp8
[BIs: 60, 966 reqs                        ] model: mistralai/voxtral-small-24b-2507  provider: mistral
[BIs: 58, 900 reqs                        ] model: deepseek/deepseek-v3.2-exp  provider: novita/fp8
[BIs: 54, 492 reqs                        ] model: deepseek/deepseek-v3.2  provider: chutes/fp8
[BIs: 54, 900 reqs                        ] model: deepseek/deepseek-chat-v3.1  provider: deepinfra/fp4
[BIs: 52, 1014 reqs, reasoning: disabled  ] model: z-ai/glm-4.7-flash  provider: deepinfra/bf16
[BIs: 51, 1530 reqs                       ] model: z-ai/glm-4.7-flash  provider: z-ai
[BIs: 51, 3001 reqs                       ] model: qwen/qwen3-8b  provider: atlas-cloud/fp8
[BIs: 49, 2999 reqs                       ] model: qwen/qwen3-8b  provider: alibaba
[BIs: 45, 807 reqs                        ] model: deepseek/deepseek-v3.1-terminus  provider: deepinfra/fp4
[BIs: 45, 939 reqs                        ] model: cohere/command-r-08-2024  provider: cohere
[BIs: 43, 3000 reqs                       ] model: meta-llama/llama-3.1-70b-instruct  provider: deepinfra/turbo
[BIs: 41, 297 reqs                        ] model: tencent/hunyuan-a13b-instruct  provider: siliconflow/fp8
[BIs: 40, 468 reqs                        ] model: deepseek/deepseek-v3.2-exp  provider: atlas-cloud/fp8
[BIs: 40, 954 reqs                        ] model: mistralai/ministral-8b-2512  provider: mistral
[BIs: 40, 1113 reqs                       ] model: google/gemma-3-27b-it  provider: parasail/fp8
[BIs: 34, 936 reqs                        ] model: mistralai/ministral-14b-2512  provider: mistral
[BIs: 33, 2997 reqs                       ] model: openai/gpt-4o-mini  provider: openai
[BIs: 32, 3000 reqs                       ] model: openai/gpt-4o-mini-2024-07-18  provider: openai
[BIs: 31, 762 reqs                        ] model: qwen/qwen3-next-80b-a3b-instruct  provider: alibaba
[BIs: 30, 789 reqs                        ] model: qwen/qwen3-vl-30b-a3b-instruct  provider: alibaba
[BIs: 29, 2988 reqs                       ] model: amazon/nova-lite-v1  provider: amazon-bedrock
[BIs: 29, 3000 reqs                       ] model: google/gemini-2.0-flash-lite-001  provider: google-ai-studio
[BIs: 29, 3000 reqs                       ] model: sao10k/l3-lunaris-8b  provider: novita/bf16
[BIs: 28, 747 reqs                        ] model: qwen/qwen3-vl-8b-instruct  provider: novita/fp8
[BIs: 28, 3000 reqs                       ] model: google/gemini-2.0-flash-lite-001  provider: google-vertex
[BIs: 27, 3000 reqs                       ] model: mistralai/ministral-8b-2512  provider: nextbit/fp8
[BIs: 26, 3000 reqs                       ] model: openai/gpt-4o-mini  provider: azure
[BIs: 25, 327 reqs                        ] model: deepseek/deepseek-v3.2  provider: akashml/fp8
[BIs: 25, 597 reqs                        ] model: meta-llama/llama-3.3-70b-instruct  provider: novita/bf16
[BIs: 25, 981 reqs                        ] model: meta-llama/llama-3-8b-instruct  provider: novita/bf16
[BIs: 24, 522 reqs                        ] model: qwen/qwq-32b  provider: siliconflow/fp8
[BIs: 24, 618 reqs, reasoning: disabled   ] model: qwen/qwen3-30b-a3b  provider: novita/fp8
[BIs: 23, 708 reqs                        ] model: meta-llama/llama-3.1-8b-instruct  provider: nebius/fp8
[BIs: 21, 807 reqs                        ] model: qwen/qwen3-coder-30b-a3b-instruct  provider: novita/fp8
[BIs: 20, 3000 reqs                       ] model: meta-llama/llama-3-8b-instruct  provider: deepinfra/bf16
[BIs: 20, 3000 reqs                       ] model: qwen/qwen-2.5-7b-instruct  provider: together/fp8
[BIs: 19, 616 reqs                        ] model: z-ai/glm-4.5-air  provider: siliconflow/fp8
[BIs: 19, 741 reqs                        ] model: mistralai/ministral-3b-2512  provider: mistral
[BIs: 19, 3000 reqs                       ] model: google/gemini-2.5-flash-lite  provider: google-vertex
[BIs: 18, 861 reqs                        ] model: qwen/qwen3-vl-30b-a3b-instruct  provider: deepinfra/fp8
[BIs: 18, 2457 reqs                       ] model: google/gemma-3-27b-it  provider: deepinfra/fp8
[BIs: 18, 3000 reqs                       ] model: mistralai/ministral-3b-2512  provider: nextbit/fp8
[BIs: 17, 294 reqs                        ] model: z-ai/glm-4.7-flash  provider: novita/bf16
[BIs: 17, 762 reqs                        ] model: mistralai/mistral-small-3.2-24b-instruct  provider: deepinfra/fp8
[BIs: 17, 5997 reqs                       ] model: qwen/qwen3.5-flash-02-23  provider: alibaba
[BIs: 16, 3000 reqs                       ] model: mistralai/ministral-14b-2512  provider: nextbit/fp8
[BIs: 15, 2970 reqs                       ] model: thedrummer/unslopnemo-12b  provider: nextbit/fp8
[BIs: 13, 3000 reqs                       ] model: meta-llama/llama-3.1-70b-instruct  provider: deepinfra/base
[BIs: 13, 3000 reqs                       ] model: qwen/qwen3.5-9b  provider: together
[BIs: 12, 459 reqs                        ] model: qwen/qwen3-30b-a3b-instruct-2507  provider: atlas-cloud/fp8
[BIs: 12, 3000 reqs, reasoning: budget=4  ] model: alibaba/tongyi-deepresearch-30b-a3b  provider: atlas-cloud/fp8
[BIs: 12, 3000 reqs                       ] model: meta-llama/llama-3.2-11b-vision-instruct  provider: deepinfra/fp8
[BIs: 12, 3000 reqs                       ] model: mistralai/mistral-small-3.2-24b-instruct  provider: parasail/bf16
[BIs: 11, 384 reqs                        ] model: mistralai/mistral-small-3.2-24b-instruct  provider: venice/fp8
[BIs: 11, 852 reqs                        ] model: meta-llama/llama-3.1-8b-instruct  provider: novita/fp8
[BIs: 11, 3000 reqs                       ] model: mistralai/mistral-small-3.1-24b-instruct  provider: cloudflare
[BIs: 10, 3000 reqs                       ] model: meta-llama/llama-3-8b-instruct  provider: together/int4
[BIs: 10, 3000 reqs                       ] model: mistralai/mistral-nemo  provider: novita/fp8
[BIs: 10, 3000 reqs                       ] model: stepfun/step-3.5-flash  provider: stepfun/fp8
[BIs: 9, 375 reqs                         ] model: qwen/qwen3-30b-a3b-thinking-2507  provider: siliconflow/fp8
[BIs: 9, 600 reqs                         ] model: stepfun/step-3.5-flash  provider: deepinfra/fp8
[BIs: 9, 2016 reqs                        ] model: google/gemma-3-27b-it  provider: novita/bf16
[BIs: 9, 3000 reqs                        ] model: qwen/qwen3-vl-32b-instruct  provider: alibaba
[BIs: 8, 600 reqs                         ] model: qwen/qwen-vl-plus  provider: alibaba
[BIs: 8, 600 reqs                         ] model: qwen/qwen3-coder-next  provider: chutes/bf16
[BIs: 8, 3000 reqs                        ] model: google/gemma-3-12b-it  provider: cloudflare
[BIs: 8, 3000 reqs                        ] model: google/gemma-3-4b-it  provider: deepinfra/bf16
[BIs: 8, 3000 reqs                        ] model: meta-llama/llama-3.1-8b-instruct  provider: deepinfra/bf16
[BIs: 8, 5997 reqs                        ] model: amazon/nova-micro-v1  provider: amazon-bedrock
[BIs: 7, 600 reqs                         ] model: google/gemini-2.0-flash-001  provider: google-vertex
[BIs: 7, 1554 reqs, reasoning: disabled   ] model: z-ai/glm-4.5-air  provider: novita/bf16
[BIs: 6, 600 reqs                         ] model: google/gemini-2.0-flash-001  provider: google-ai-studio
[BIs: 6, 600 reqs                         ] model: meta-llama/llama-4-maverick  provider: deepinfra/base
[BIs: 6, 3000 reqs                        ] model: google/gemma-3-12b-it  provider: deepinfra/bf16
[BIs: 6, 6034 reqs                        ] model: qwen/qwen3-coder-next  provider: ionstream/fp8
[BIs: 5, 3000 reqs                        ] model: qwen/qwen3-235b-a22b-2507  provider: wandb/bf16
[BIs: 5, 3000 reqs                        ] model: sao10k/l3-lunaris-8b  provider: deepinfra/turbo
[BIs: 5, 6000 reqs                        ] model: meta-llama/llama-3.2-1b-instruct  provider: cloudflare
[BIs: 4, 6000 reqs                        ] model: liquid/lfm-2-24b-a2b  provider: together
```

```
[BIs: 4, 6000 reqs                     ] model: x-ai/grok-3-mini  provider: xai
[BIs: 3, 6018 reqs                     ] model: x-ai/grok-3-mini-beta  provider: xai
[BIs: 2, 6000 reqs                     ] model: meta-llama/llama-3.1-8b-instruct  provider: friendli
[BIs: 2, 6000 reqs                     ] model: meta-llama/llama-3.2-3b-instruct  provider: cloudflare
[BIs: 2, 6000 reqs                     ] model: mistralai/mistral-nemo  provider: deepinfra/fp8
[BIs: 2, 6000 reqs                     ] model: reka/reka-edge  provider: reka/bf16
[BIs: 1, 3477 reqs, reasoning: budget=256] model: qwen/qwen3-32b  provider: chutes/bf16
[BIs: 1, 6000 reqs                     ] model: microsoft/phi-4  provider: nextbit/int4
[BIs: 1, 6000 reqs, reasoning: budget=256] model: qwen/qwen3-32b  provider: groq
[BIs: 1, 6042 reqs                     ] model: qwen/qwen3-next-80b-a3b-thinking  provider: alibaba
[BIs: 0, 5944 reqs                     ] model: deepseek/deepseek-chat-v3.1  provider: sambanova/high-throughput
[BIs: 0, 6000 reqs                     ] model: arcee-ai/trinity-mini  provider: clarifai/bf16
[BIs: 0, 6000 reqs                     ] model: deepseek/deepseek-v3.2  provider: deepseek
[BIs: 0, 6000 reqs                     ] model: google/gemini-2.5-flash-lite  provider: google-ai-studio
[BIs: 0, 6000 reqs                     ] model: google/gemini-2.5-flash-lite-preview-09-2025  provider: google-ai-studio
[BIs: 0, 6000 reqs                     ] model: google/gemini-2.5-flash-lite-preview-09-2025  provider: google-vertex
[BIs: 0, 6000 reqs                     ] model: google/gemma-2-9b-it  provider: nebius/fast
[BIs: 0, 6000 reqs                     ] model: liquid/lfm2-8b-a1b  provider: liquid
[BIs: 0, 6000 reqs                     ] model: meta-llama/llama-3.1-8b-instruct  provider: cerebras/fp16
[BIs: 0, 6000 reqs                     ] model: meta-llama/llama-3.1-8b-instruct  provider: sambanova/bf16
[BIs: 0, 6000 reqs                     ] model: meta-llama/llama-3.3-70b-instruct  provider: nebius/fp8
[BIs: 0, 6000 reqs                     ] model: microsoft/phi-4  provider: deepinfra/bf16
[BIs: 0, 6000 reqs                     ] model: mistralai/devstral-small  provider: mistral
[BIs: 0, 6000 reqs                     ] model: mistralai/mistral-7b-instruct-v0.1  provider: cloudflare
[BIs: 0, 6000 reqs                     ] model: nousresearch/hermes-2-pro-llama-3-8b  provider: novita/fp16
[BIs: 0, 6000 reqs                     ] model: nousresearch/hermes-3-llama-3.1-70b  provider: deepinfra/fp8
[BIs: 0, 6000 reqs                     ] model: nvidia/nemotron-nano-12b-v2-v1  provider: deepinfra/fp8
[BIs: 0, 6000 reqs                     ] model: nvidia/nemotron-nano-9b-v2  provider: deepinfra/fp16
[BIs: 0, 6000 reqs                     ] model: qwen/qwen-2.5-72b-instruct  provider: novita/bf16
[BIs: 0, 6000 reqs, reasoning: budget=1  ] model: qwen/qwen3-14b  provider: deepinfra/fp8
[BIs: 0, 6000 reqs, reasoning: budget=128] model: qwen/qwen3-14b  provider: nextbit/int4
[BIs: 0, 6000 reqs, reasoning: budget=256] model: qwen/qwen3-30b-a3b  provider: deepinfra/fp8
[BIs: 0, 6000 reqs, reasoning: budget=512] model: qwen/qwen3-30b-a3b  provider: friendli
[BIs: 0, 6000 reqs                     ] model: qwen/qwen3-30b-a3b-instruct-2507  provider: wandb/bf16
[BIs: 0, 6000 reqs                     ] model: qwen/qwen3-30b-a3b-thinking-2507  provider: nebius/fp8
[BIs: 0, 6000 reqs, reasoning: budget=256] model: qwen/qwen3-32b  provider: deepinfra/fp8
[BIs: 0, 6000 reqs                     ] model: qwen/qwen3-vl-8b-instruct  provider: alibaba
[BIs: 0 (hidden reasoning)             ] model: openai/gpt-oss-20b  provider: amazon-bedrock
[BIs: 0 (hidden reasoning)             ] model: openai/gpt-oss-120b  provider: amazon-bedrock
```

## E.4. Endpoints used for B3IT tracking

The following 93 endpoints were searched for Border Inputs at $T = 0$ with up to 5,000 single-token prompts per endpoint, sampled 3 times each. For each, we report the number of BIs found and the total number of API requests sent during the search. Endpoints with at least one BI were subsequently monitored continuously over 23 days.

```
[BIs: 61, 1720 reqs ]  model: mistralai/mistral-nemo  provider: mistral
[BIs: 60, 540 reqs  ]  model: mistralai/mistral-nemo  provider: azure
[BIs: 60, 780 reqs  ]  model: deepseek/deepseek-v3.2  provider: parasail/fp8
[BIs: 60, 880 reqs  ]  model: deepseek/deepseek-chat-v3-0324  provider: fireworks
[BIs: 60, 960 reqs  ]  model: relace/relace-search  provider: relace/bf16
[BIs: 60, 1120 reqs ]  model: mistralai/ministral-3b  provider: mistral
[BIs: 60, 2440 reqs ]  model: qwen/qwen3-vl-30b-a3b-instruct  provider: fireworks
[BIs: 60, 2620 reqs ]  model: google/gemma-3-12b-it  provider: chutes/bf16
[BIs: 60, 4980 reqs ]  model: qwen/qwen3-235b-a22b-2507  provider: wandb/bf16
[BIs: 60, 7260 reqs ]  model: mistralai/mistral-nemo  provider: chutes/bf16
[BIs: 60, 10320 reqs]  model: google/gemma-3-4b-it  provider: deepinfra/bf16
[BIs: 60, 10780 reqs]  model: deepseek/deepseek-chat-v3.1  provider: sambanova/fp8
[BIs: 60, 12580 reqs]  model: qwen/qwen3-235b-a22b-2507  provider: crusoe/bf16
[BIs: 60, 14860 reqs]  model: qwen/qwen-2.5-coder-32b-instruct  provider: chutes/fp8
[BIs: 59, 1320 reqs ]  model: openai/gpt-4o  provider: openai
[BIs: 59, 1340 reqs ]  model: deepseek/deepseek-chat-v3-0324  provider: hyperbolic/fp8
[BIs: 59, 2340 reqs ]  model: kwaipilot/kat-coder-pro  provider: streamlake/fp16
[BIs: 59, 5880 reqs ]  model: openai/gpt-4o-mini  provider: openai
[BIs: 59, 6740 reqs ]  model: mistralai/devstral-small-2505  provider: deepinfra/bf16
[BIs: 59, 10340 reqs]  model: mistralai/mistral-small-3.2-24b-instruct  provider: parasail/bf16
[BIs: 58, 580 reqs  ]  model: deepseek/deepseek-chat-v3-0324  provider: atlas-cloud/fp8
[BIs: 58, 1120 reqs ]  model: cohere/command-r-08-2024  provider: cohere
[BIs: 58, 1340 reqs ]  model: mistralai/ministral-8b-2512  provider: mistral
[BIs: 58, 1360 reqs ]  model: mistralai/pixtral-12b  provider: mistral
[BIs: 58, 4520 reqs ]  model: moonshotai/kimi-k2-0905:exacto  provider: moonshotai
[BIs: 58, 6920 reqs ]  model: google/gemma-3-27b-it  provider: chutes/bf16
[BIs: 58, 7480 reqs ]  model: openai/gpt-4o-mini  provider: azure
[BIs: 58, 11680 reqs]  model: amazon/nova-micro-v1  provider: amazon-bedrock
[BIs: 57, 1200 reqs ]  model: mistralai/ministral-3b-2512  provider: mistral
[BIs: 57, 1400 reqs ]  model: deepseek/deepseek-v3.2  provider: phala
[BIs: 57, 2040 reqs ]  model: google/gemma-3-12b-it  provider: crusoe/bf16
[BIs: 56, 1500 reqs ]  model: mistralai/ministral-8b  provider: mistral
[BIs: 56, 10480 reqs]  model: amazon/nova-lite-v1  provider: amazon-bedrock
[BIs: 56, 15000 reqs]  model: microsoft/phi-4  provider: deepinfra/bf16
[BIs: 52, 3620 reqs ]  model: inflection/inflection-3-pi  provider: inflection
[BIs: 50, 4720 reqs ]  model: anthropic/claude-sonnet-4.5  provider: anthropic
[BIs: 43, 1740 reqs ]  model: meta-llama/llama-guard-2-8b  provider: together
[BIs: 42, 6040 reqs ]  model: google/gemma-3-12b-it  provider: novita/bf16
[BIs: 40, 15000 reqs]  model: google/gemma-3-12b-it  provider: deepinfra/bf16
[BIs: 36, 15000 reqs]  model: anthropic/claude-haiku-4.5  provider: anthropic
[BIs: 36, 15000 reqs]  model: microsoft/phi-4-multimodal-instruct  provider: deepinfra/bf16
[BIs: 35, 8840 reqs ]  model: mistralai/mistral-nemo  provider: deepinfra/fp8
```

```
[BIs: 28, 4920 reqs ]  model: qwen/qwen-2.5-72b-instruct  provider: hyperbolic/bf16
[BIs: 26, 15000 reqs]  model: mistralai/mistral-7b-instruct-v0.3  provider: together
[BIs: 25, 15000 reqs]  model: nousresearch/hermes-2-pro-llama-3-8b  provider: nextbit/int4
[BIs: 23, 600 reqs ]  model: deepseek/deepseek-chat-v3-0324  provider: baseten/fp8
[BIs: 21, 15000 reqs]  model: qwen/qwen3-235b-a22b-2507  provider: cerebras
[BIs: 19, 15000 reqs]  model: qwen/qwen3-vl-235b-a22b-instruct  provider: fireworks
[BIs: 14, 6060 reqs ]  model: alibaba/tongyi-deepresearch-30b-a3b  provider: ncompass/bf16
[BIs: 11, 2800 reqs ]  model: google/gemma-3-27b-it  provider: phala
[BIs: 10, 15000 reqs]  model: google/gemma-3-4b-it  provider: chutes
[BIs: 7, 15000 reqs ]  model: mistralai/devstral-medium  provider: mistral
[BIs: 1, 15000 reqs ]  model: microsoft/phi-4  provider: nextbit/int4
[BIs: 0, 0 reqs     ]  model: thudm/glm-4.1v-9b-thinking  provider: novita/bf16
[BIs: 0, 120 reqs   ]  model: qwen/qwen3-coder  provider: baseten/fp8
[BIs: 0, 8920 reqs  ]  model: qwen/qwen3-32b  provider: friendli
[BIs: 0, 10240 reqs ]  model: qwen/qwen3-30b-a3b  provider: friendli
[BIs: 0, 14980 reqs ]  model: anthropic/claude-sonnet-4.5  provider: amazon-bedrock
[BIs: 0, 14980 reqs ]  model: baidu/ernie-4.5-21b-a3b  provider: novita/bf16
[BIs: 0, 14980 reqs ]  model: cohere/command-r7b-12-2024  provider: cohere
[BIs: 0, 14980 reqs ]  model: z-ai/glm-4.6:exacto  provider: z-ai
[BIs: 0, 15000 reqs ]  model: deepseek/deepseek-r1-distill-llama-70b  provider: chutes/bf16
[BIs: 0, 15000 reqs ]  model: deepseek/deepseek-v3.2  provider: atlas-cloud/fp8
[BIs: 0, 15000 reqs ]  model: deepseek/deepseek-v3.2  provider: deepseek
[BIs: 0, 15000 reqs ]  model: google/gemini-2.5-flash-lite-preview-09-2025  provider: google-ai-studio
[BIs: 0, 15000 reqs ]  model: google/gemini-2.5-flash-lite-preview-09-2025  provider: google-vertex
[BIs: 0, 15000 reqs ]  model: google/gemini-3-flash-preview  provider: google-ai-studio
[BIs: 0, 15000 reqs ]  model: google/gemini-3-flash-preview  provider: google-vertex
[BIs: 0, 15000 reqs ]  model: google/gemma-2-9b-it  provider: nebius/fast
[BIs: 0, 15000 reqs ]  model: google/gemma-3-27b-it  provider: ncompass/fp8
[BIs: 0, 15000 reqs ]  model: liquid/lfm-2.2-6b  provider: liquid
[BIs: 0, 15000 reqs ]  model: liquid/lfm2-8b-a1b  provider: liquid
[BIs: 0, 15000 reqs ]  model: meta-llama/llama-3.1-405b  provider: hyperbolic/bf16
[BIs: 0, 15000 reqs ]  model: mistralai/mistral-7b-instruct-v0.1  provider: cloudflare
[BIs: 0, 15000 reqs ]  model: moonshotai/kimi-k2-thinking  provider: moonshotai/int4
[BIs: 0, 15000 reqs ]  model: nousresearch/hermes-2-pro-llama-3-8b  provider: novita/fp16
[BIs: 0, 15000 reqs ]  model: openai/gpt-oss-120b  provider: amazon-bedrock
[BIs: 0, 15000 reqs ]  model: openai/gpt-oss-20b  provider: amazon-bedrock
[BIs: 0, 15000 reqs ]  model: prime-intellect/intellect-3  provider: nebius/fp8
[BIs: 0, 15000 reqs ]  model: qwen/qwen3-14b  provider: chutes/bf16
[BIs: 0, 15000 reqs ]  model: qwen/qwen3-14b  provider: nextbit/int4
[BIs: 0, 15000 reqs ]  model: qwen/qwen3-14b  provider: deepinfra/fp8
[BIs: 0, 15000 reqs ]  model: qwen/qwen3-30b-a3b-thinking-2507  provider: cloudflare
[BIs: 0, 15000 reqs ]  model: qwen/qwen3-32b  provider: nebius/base
[BIs: 0, 15000 reqs ]  model: qwen/qwen3-32b  provider: groq
[BIs: 0, 15000 reqs ]  model: qwen/qwen3-32b  provider: cerebras
[BIs: 0, 15000 reqs ]  model: qwen/qwen3-32b  provider: sambanova
[BIs: 0, 15000 reqs ]  model: qwen/qwen3-8b  provider: fireworks
[BIs: 0, 15000 reqs ]  model: z-ai/glm-4-32b  provider: z-ai
[BIs: 0, 15000 reqs ]  model: z-ai/glm-4.5  provider: wandb/bf16
[BIs: 0, 15000 reqs ]  model: z-ai/glm-4.5-air  provider: chutes/bf16
[BIs: 0, 15000 reqs ]  model: z-ai/glm-4.6  provider: mancer/fp8
[BIs: 0, 15000 reqs ]  model: z-ai/glm-4.7  provider: mancer/fp8
```

### E.5. Selecting minimum-length prompts

We are looking for prompt strings that are encoded into a minimal number of tokens. To find them, we download 11 tokenizers, representing 47 models from the Deepseek, Qwen, Gemma, LLaMA, Phi, Mistral, and GPT-OSS series. For each tokenizer, we select unique decoded tokens from its vocabulary, replacing the special characters U+0120 and U+2581 with a space, skipping special tokens and tokens that get encoded into more than 2 token IDs (which can happen due to weird representation of partial UTF-8 characters, itself due to BPE tokenization). For models with an unknown tokenizer (e.g. the Claude model series), we use a proxy tokenizer where the prompts are in order of decreasing frequency among the known tokenizers (prompts that many tokenizers encode as a single token are more likely to be encoded as a single token by an unknown tokenizer).

### E.6. TinyChange experimental parameters and methodology

For all methods, models, variants, and method parameters, all prompts were initially sampled 5,000 times each. 1,000 test statistic were subsequently generated, using random sampling in the prompts (where appropriate) and in the samples for each prompt. Unless otherwise specified, the number of detection samples is 10. Error bars in Figures 3 and 7 come from 250 bootstraps holding everything fixed except the test statistics for each condition, which are resampled with replacement. 95% confidence intervals are extracted from these bootstraps.

### E.7. Parameter sweeps

Figure 6 shows the hyperparameters that were varied for each of the baselines.

For the TinyChange difficulty scale plots, we kept the following hyperparameters, always keeping 50 reference sam-

ples/prompt:

- B3IT: 5 prompts, 3 detection samples/prompt (sweep: see Figure 1)

- MET: 25 prompts, 5 output tokens (10 detection samples/prompt)

- MMLU-ALG: 100 prompts (10 detection samples/prompt)

- LT: 10 detection samples/prompt (1 prompt)

### E.8. Misc

**Note on OLMo.**    For OLMo-2-1124-7B in our *in vitro* experiments, the first token was the same for all inputs. We therefore generated two tokens and retained the second. This workaround slightly increases cost, but preserves the method.

### E.9. Full TinyChange difficulty scales

Figure 7 shows four more difficulty scales, in addition to the fine-tuning one presented in the main paper.

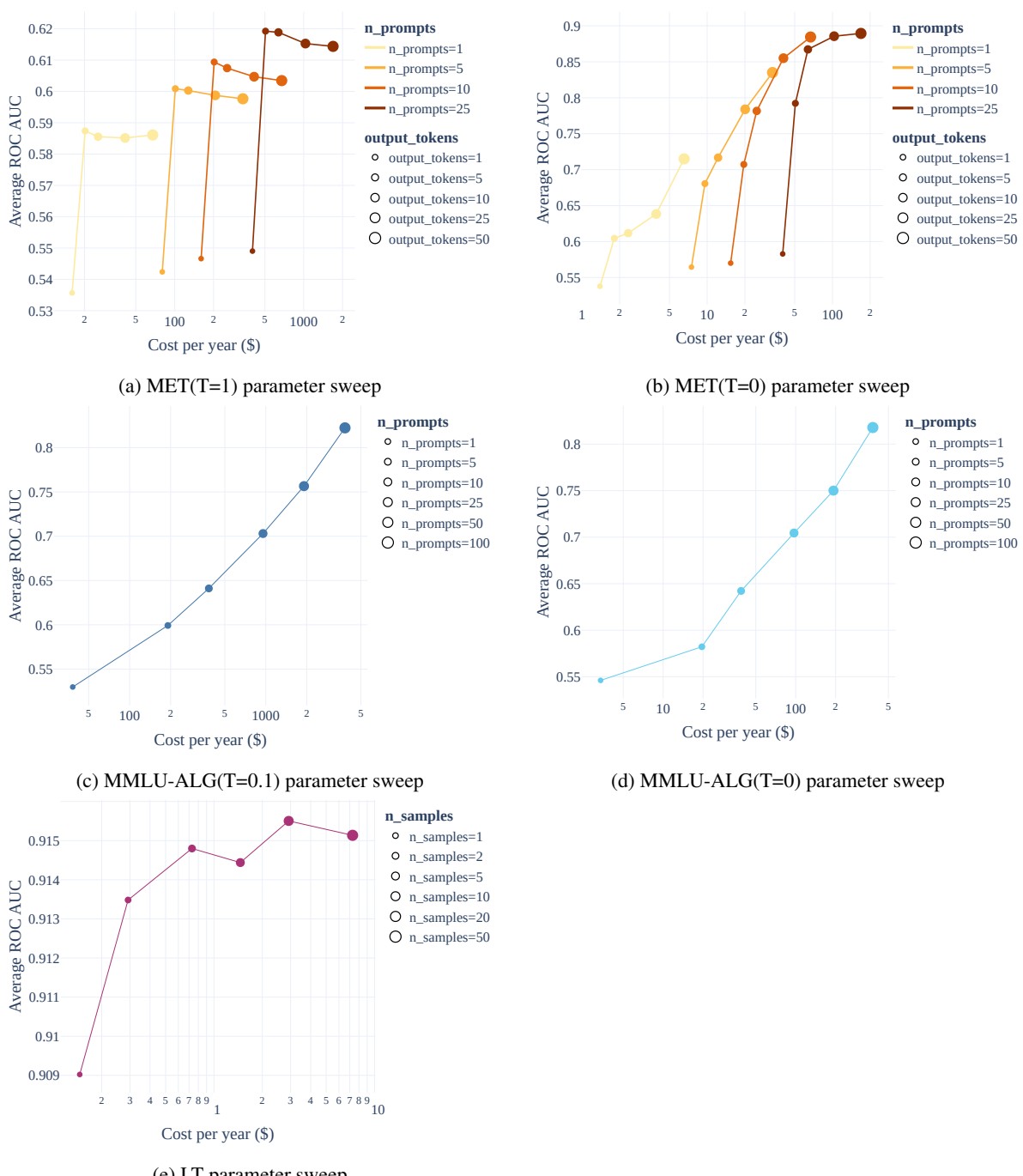

(a) MET(T=1) parameter sweep

(b) MET(T=0) parameter sweep

(c) MMLU-ALG(T=0.1) parameter sweep

(d) MMLU-ALG(T=0) parameter sweep

(e) LT parameter sweep

*Figure 6.* Hyperparameter sweeps for the baseline methods on the TinyChange fine-tuning difficulty scale.

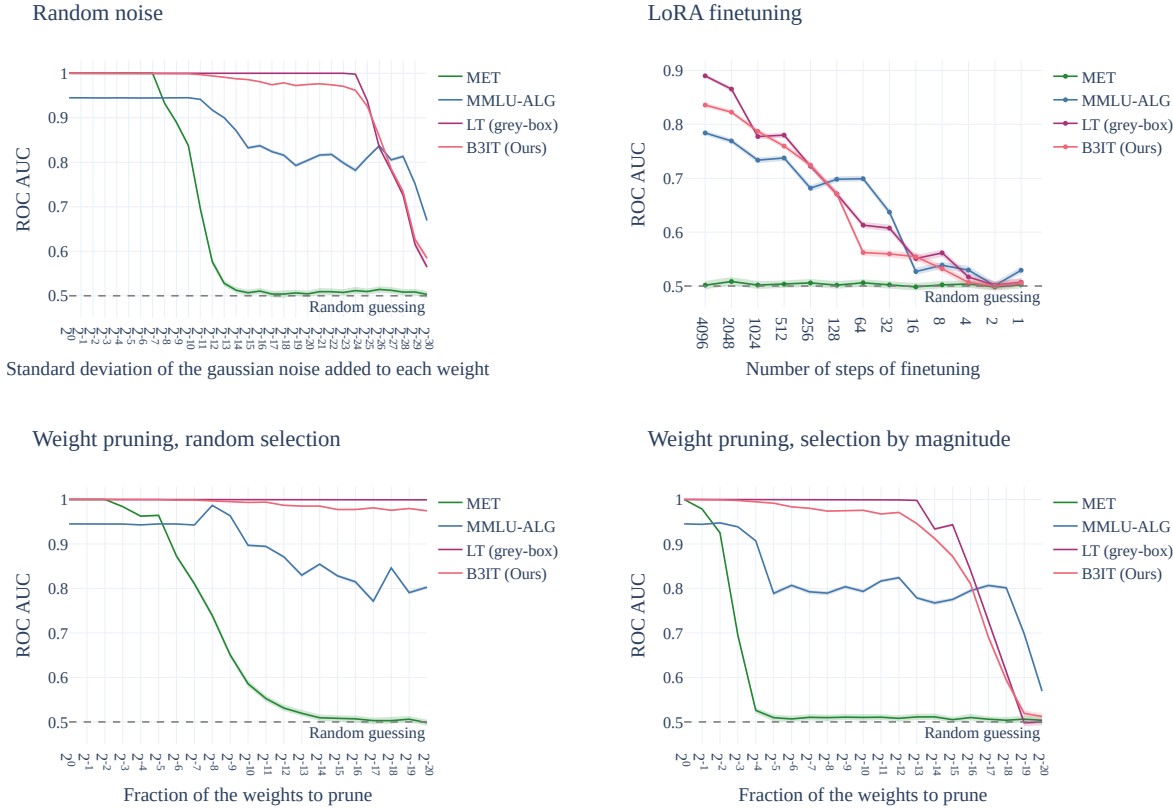

*Figure 7.* Detection performance on the TinyChange difficulty scales, for various perturbations.

