# OpenReview forum: "Token-Efficient Change Detection in LLM APIs"
_ICML.cc/2026/Conference — ICML 2026 regular_

### Official Review · Reviewer_mhWu · 2026-03-13

**Soundness:** 3
**Presentation:** 3
**Significance:** 3
**Originality:** 3
**Overall Recommendation:** 4
**Confidence:** 3

**Summary:**

This work focuses on the problem of detecting model changes for LLM APIs from a strict black-box perspective, where only output tokens are available. It provides a theory for change detectability at low temperature, which proves a phase transition for Border Inputs (BIs) whose top-two logits are tied. This allows for high query efficiency testing. Following from this theory, the authors propose a simple method for change detection called B3IT, which consists of a lightweight procedure for finding BIs and then tracking uniform support changes for a few output tokens. Experiments on the TinyChange benchmark and 93 live APIs show that B3IT results in significant cost savings, around 30x compared to the previous black-box methods while reaching the accuracy of a grey-box logprob-based baseline.

**Compliance With Llm Reviewing Policy:**

Affirmed.

**Key Questions For Authors:**

1. How much does it depend on other decoding parameters, such as top-p, top-k, min-p, repetition penalty, etc., instead of temperature? Do you control or detect these parameters on commercial endpoints?
2. Your cost model is based on hourly monitoring with certain price points. How does it change if rate limits are more strict, if token prices are different, or if endpoints have higher latency that requires retries or timeouts?
3. Can you perform an ablation on the BI search strategy, e.g., changing prompts to deliberately try to reach near-border regions, and how it influences query budgets?

**Limitations:**

yes

**Strengths And Weaknesses:**

Strengths
1. The paper includes comparisons against strong black-box and grey-box baselines (MET, MMLU-ALG, LT), as well as temperature variants to ensure fair conditions in the low-T regime emphasized by the theory.
2. The method B3IT is simple to understand and implement. They also stated error guarantees with practical guidance (e.g., m=3, typical k in {2,3}). Codes are also released and are clear to read.
3. I agree that this could be a real problem and a growing need for cost-effective, black-box monitoring of LLM APIs subject to frequent unannounced changes. This work provides new insights on this direction.

Weaknesses:
1. The study assumes that the temperature is nearly zero and that the averages are nearly equal; in theory, the exact average must be zero; however, the study does not provide a formal model for quantifying the likelihood that this occurs due to uncertainty.
2. The BI prevalence is briefly mentioned with 62% of the endpoints at T=0, around 80% for T>0 in the text in section 5.2.2. However, it is not supported by a stability analysis over time, load, or updates.

---

> ### Author Rebuttal · Authors · 2026-03-31
>
> We thank the reviewer for the valuable feedback, and for appreciating the operational setup and the theoretical guidance of our work. Regarding the weaknesses and questions mentioned in the review:
>
> ### W1: Limited theoretical guarantees, only low temperature
>
> We note that Theorem 3.1 and Lemma 3.2 hold for any temperature. These results, in addition to being novel by themselves, enable us to prove Theorem 3.3, showing that detectability is maximal for Border Inputs at low temperature. Low temperature is thus the optimal regime for change detection, not a condition for our main results to hold. As $T=0$ can be set for all LLM APIs we are aware of, this becomes the basis of the B3IT technique.
>
> ### W2: Cost and robustness of Border Input discovery
>
> At the end of page 7, we report that Border Input discovery costs 0.0045 USD per successful endpoint on average. Figure 4 also shows how many queries are typically required to identify Border Inputs: for instance, at $T=0$, 50% of endpoints require fewer than 300 requests (100 prompts) per successful BI.
>
> For more discussion on BI search strategies, please see our answer to mhWu (Q3). On the fraction of endpoints where we can easily identify BIs, please see our answer to Cccd (W1,W3).
>
> ### W3: Evaluation scope
>
> We validate B3IT extensively:
>
> * in-vitro on 9 LLMs from 7 model families, on 5 difficulty scales from the TinyChange benchmark (Figure 3, Figure 7);
> * in-vivo on 93 endpoints (64 models, 38 providers) for the BI prevalence study, and 54 endpoints for continuous monitoring.
>
> We now have results for longer-term monitoring (see our [updated Figure 5](https://anonymous.4open.science/r/token-efficient-change-detection-llm-apis-561D/b3it_monitoring/assets/updated_fig5.pdf)). Multiple undisclosed changes were identified. One of the changes was confirmed as a model swap from the provider's [changelog](https://docs.together.ai/docs/changelog#jan-29).
>
> ### W4: Practical deployment considerations
>
> We send queries every 24h, but determining a frequency is up to the auditor, depending on their budget.
>
> Inference non-determinism should already be covered by our in-vivo experiments. We are unsure what the reviewer refers to by "noisy outputs" beyond this, and would welcome clarification so we can address it properly.
>
> Regarding the stochastic decoding settings, please see our response to Q2 below.
>
> Our source code can also serve as an example of a practical deployment.
>
> ### Q1: Cost of Border Input Discovery
>
> Please see our answer to W2.
>
> ### Q2: Robustness to decoding settings
>
> Our method focuses on APIs where the temperature can be set by the user, which represents all LLM APIs that we are aware of. Decoding parameters such as top-k, top-p or min-p are also set by the user, so we believe that it would be outside of the scope of this study to be robust to these settings.
>
> ### Q3: Types of detectable changes
>
> Our evaluation on the TinyChange benchmark (Figures 3 and 7) show that detection is easier for larger changes, and harder for smaller changes. Regarding architectural changes, our in-vivo experiment detected the model swap from Mistral-7B-Instruct-v0.3 to Ministral-3-14B-Instruct-2512 unambiguously, with the mean TV distance jumping from $\approx 0.05$ to $1$ (see W3).
>
> ### Q4: Generalization across models
>
> This is a very interesting question. As we already had the data, we tested whether BIs found for an endpoint are more likely to be BIs for another endpoint.
>
> We ran the Cochran–Mantel–Haenszel test.
>
> | Pair type                 | Models | Endpoints | CMH Odds Ratio (95% CI) | *p*     |
> | ------------------------- | ------ | --------- | ----------------------- | ------- |
> | Same model, diff provider | 28     | 86        | 2.40 (2.12, 2.72)       | 1.4e-49 |
> | Different model           | 55     | 104       | 1.07 (1.04, 1.11)       | 1.2e-04 |
>
> BIs are indeed correlated for the same model on different providers: a BI on an endpoint is 2.4x more likely to be a BI on another same-model endpoint than the base rate. This correlation is expected since the model is supposed to be the same.
>
> For different models, the effect is much smaller and close to independence: a BI on a model is only 1.07x more likely to be a BI on a different model. The fact that BIs don't generalize across models relates to their effectiveness as change detectors: by design, they are very likely to break down on a model change, which also means they are unlikely to generalize across models.
>
> We will add these results to the paper.
>
> ### Q5: Practical deployment considerations
>
> The choice of how often to run B3IT will be up to the auditor, with their specific budget and requirements. Given its low cost, nothing prevents B3IT from being run much more often than we did (every 24h). As we request a single token of output, cost is fully predictable. The auditor may also use our Figure 1 to decide on the number of prompts and samples per prompt that will achieve the best performance given their budget.

---

> > ### Author Rebuttal · Reviewer_mhWu · 2026-04-03
> >
> > The rebuttal appears to address a different reviewer's questions. My Q2 about cost model variations and Q3 about BI search ablation were not addressed. The rebuttal directs me to "see our answer to mhWu (Q3)" for the BI search ablation but I am mhWu. Given this, I maintain my score.

---

> > > ### Author Response · Authors · 2026-04-04
> > >
> > > We sincerely apologize, our original response mistakenly addressed questions from a different reviewer. Please find below the rebuttal we had originally prepared addressing your specific points. We understand the timing is not ideal, but we hope these clarifications are still helpful for your assessment.
> > >
> > > We thank the reviewer for the valuable feedback, for having appreciated the work motivation and extensive experimentation, and for having a look at our source code. Regarding the weaknesses and questions mentioned in the review:
> > >
> > > ### W1: Formal model of the likelihood of tied top tokens
> > >
> > > Our statistical analysis relies on the standard assumption that logprobs are continuous, which is necessary for the asymptotic analysis and yields our main results on the power of border inputs. A notable consequence of this assumption is that tied top tokens have probability zero, so BIs should theoretically never exist.
> > >
> > > In practice, however, logprobs have discrete support due to limited floating-point precision and inference-time non-determinism (as discussed at the beginning of Section 5, line 286), which enables our theoretical analysis to be applied in practice.
> > >
> > > Providing a formal model of either of these factors would be challenging. Analyzing the limited floating-point precision would require a model of the distribution of the top logprobs, which depends on the prompt, training data and architecture. The inference non-determinism depends on factors such as the GPU load, specific hardware used, and software stack, and the logprob distributions display great variability between endpoints (see e.g. Figures 6 to 14 in [1]). Incorporating such uncertainty in our formal model, if tractable at all, would add substantial complexity. We instead treat BI prevalence as an empirical question: Figure 4 from the in-vivo study shows that BIs are readily found on the majority of commercial endpoints.
> > >
> > > [1] Chauvin et al, 2026: Log Probability Tracking of LLM APIs
> > >
> > > ### W2: No stability analysis over time, load, or updates
> > >
> > > Informally, we consider our 93 studied endpoints as representative samples from a space of possible endpoints, which already incorporates the notions of time, load and updates. We believe the discoverability of BIs depends mostly on the magnitude of the inference non-determinism, which typically doesn't change over time or updates.
> > >
> > > We pursued data collection after the submission and will update the paper with these new results (see our [updated Figure 5](https://anonymous.4open.science/r/token-efficient-change-detection-llm-apis-561D/b3it_monitoring/assets/updated_fig5.pdf)). These results led to the detection of changes, some of which are confirmed by providers.
> > >
> > > ### Q1: Other decoding parameters
> > >
> > > Our method focuses on APIs where the temperature can be set by the user, which represents all commercial endpoints that we are aware of. For these endpoints, decoding parameters such as top-k, top-p, min-p or repetition penalty are also set by the user. In practice, default values don't interfere with the method.
> > >
> > > ### Q2: Endpoint rate limits / latency / prices
> > >
> > > * if rate limits are more strict: our method requires few queries. An API that wouldn't allow 15 queries per day or per hour would quickly be in financial trouble. That being said, we do sometimes run into rate limits (HTTP code 429) on OpenRouter, due to the traffic from all OpenRouter clients being aggregated. In these cases, we retry with jitter and exponential backoff, but this doesn't affect the cost, since HTTP 429 responses are not billed.
> > > * if token prices are different, the cost is still predictable, as it scales with the input and output token prices.
> > > * endpoints with a very high latency (e.g. more than 60s to return a response) could indeed alter the cost if the client timed out before receiving the response, but the response had already been generated. We did not observe endpoints with such abnormally high response latencies, especially given the fast generation time for our requests (a single token of output).
> > >
> > > ### Q3: Other BI search strategy
> > >
> > > We would like to point out that the BI search procedure is already extremely cheap, as it costs $0.0045 per successful endpoint on average, as reported at the end of page 7.
> > >
> > > Nonetheless, this is a very interesting question. A first idea could be to see if some heuristics could be used by leveraging common BIs. We tested whether BIs found for an endpoint are more likely to be BIs for another endpoint, considering both different providers for the same model, and different models. Please see our answer to cuVM for the numerical details (Q4). It turns out that except for different providers of the same model, BIs don't transfer to other models.
> > >
> > > Note that the black box setting seems to prevent any direct optimization method to find border inputs, and that we proved the optimality of our number of requests per candidate prompt (3) in Appendix C (page 23).

---

### Official Review · Reviewer_kgpd · 2026-03-13

**Soundness:** 3
**Presentation:** 1
**Significance:** 2
**Originality:** 3
**Overall Recommendation:** 2
**Confidence:** 2

**Summary:**

The paper illustrates a strategy to detect changes in trained LLM. The idea is to focus the analysis on "border inputs" which provide after multiple queries, outputs that are significantly varying. The detection scheme is designed for the tokens returned from this prompts, and is based on the outputs comparison at token level. While I like the general idea, I believe authors should provide more elements to translate this method into practice.

**Compliance With Llm Reviewing Policy:**

Affirmed.

**Final Justification:**

I confirm my original rating of this manuscript, that is too far from acceptance in the current format

**Key Questions For Authors:**

What does it mean at line 138 that \eps -> 0 too quickly? with respect to what is too quick? I assume the change is fixed and there is no time evolution in the data.

I am rather skeptical that the change assumption \theta \to \theta + \eps h is representative of the mentioned scenarios like quantization/fine-tuning (the considered settings). No discussion is provided. Being this assumption underpinning all the theoretical developments, it can make vacuous all the theoretical results.

At line 143-145 authors claim that the natural framework for the a meaningful testing problem si when the change in parameters is meaningful when the parameters change in a unperceivable manner. I am rater puzzle of this claim. Are we sure we that changes claimed in the introduction are unperceivable in hundred million dimensional space? This should be backed by empirical arguments and some comments as well. Otherwise there is the risk that theoretical results are not relevant to address the problem outlined in the introduction. Fine tuning and quantization are perhaps unperceivable but not because \eps \approx 0, but simply because due to the curse of dimensionality, the change detectability vanishes. I encourage authors to substantiate their claim in the actual problem addressed, otherwise theoretical derivations are vacuous.

The role of temperature seems key, but the authors did not mention what this actually corresponds to and how to regulate that (is that T or \tau?). Is that fixed after training? I think the authors should better discuss how in practice we can speculate on this parameter and what does it means this parameter is "low".

While I appreciate the "Takeaway paragraphs", Algorithms describing both the Initialization and the Detection phases are needed. I believe authors simply focused on the theoretical developments and leave the "how to implement" the described change detection test vague. This severely prevents reproducibility of the method.

**Limitations:**

Yes

**Strengths And Weaknesses:**

Soundness: the idea is nice and backed by formal analysis.

Presentation: I appreciate the effort of translating theoretical results into practice. The main body of the article however should also translate these into algorithmic solution and this aspect is completely missing. Probably this is simply because this did not fit in the page limit. However, I believe this is an important aspect for an ICML paper. Overall, the narrative is rather poor, the rationale leading statistical developments is missing (see for instance 116 - 121). A good conference paper should not be packed of formulas, but able to clearly outline and describe ideas to a relatively broad audience.

Significant: again, the idea is good and seem to work in experiments. However, the major limitation is that it is not described how to turn tis into an algorithmic detection scheme.

Originality: to the best of my knowledge the solution is original

---

> ### Author Rebuttal · Authors · 2026-03-31
>
> Thank you for your valuable feedback, and for appreciating the soundness of our proposal. Regarding weaknesses and questions:
>
> ### W1, W2, Q5: Description of the algorithm
>
> The algorithm is described in Sections 4.1 and 4.2. Its pseudocode is also included in Algorithms 1 and 2 in the Appendix (page 24), and referenced at the end of Section 4 (line 276). Regarding reproducibility, we also open-sourced a full implementation, which we referenced in a footnote at the end of the introduction (line 107).
>
> We also took care to make our analysis accessible to a broad audience, despite its complexity. In what way do you think that the rationale leading to statistical developments is missing, lines 116-121 or elsewhere?
>
> ### Q1: Meaning of "epsilon going to 0 too quickly"
>
> The rate is with respect to the sample size $n$ (line 113). To see why this regime is the right one: for fixed $\epsilon$, the problem is intractable due to the highly non-convex nature of LLMs. Our key insight is that letting $\epsilon \to 0$ allows us to relate the problem to differential quantities of the model, making it tractable. However, the rate matters critically: if $\epsilon \to 0$ faster than $1/\sqrt{n}$, the change becomes statistically undetectable regardless of the test used; if slower, detection becomes trivial. The $1/\sqrt{n}$ rate is the unique regime where the problem is neither impossible nor trivial, this is precisely what the LAN framework captures. Our contribution is to show that this statistical framework applies to LLM change detection, and that it reveals which quantities, Jacobian, Fisher information, temperature, govern detection difficulty. We will clarify this in the revised version.
>
> ### Q2: Representativeness of the theta + epsilon h assumption
>
> Any change that preserves the architecture can be written as $\theta_0 + \epsilon h$ (with $h$ a unit direction and $\epsilon$ a magnitude), so the additive form itself is not restrictive. The key assumption is that $\epsilon$ is small (specifically, $\epsilon_n = s/\sqrt{n}$ in the LAN regime), which is what makes the analysis tractable and yields our main result: that border inputs at low temperature are powerful probes for change detection.
>
> Our experiments then show that B3IT remains an effective method well beyond this regime. As shown in Figures 3 and 7, B3IT achieves strong detection across fine-tuning up to 4,096 steps, pruning up to 100% of weights, and Gaussian noise with up to $\sigma = 1$. Our in-vivo study (see our [updated Figure 5](https://anonymous.4open.science/r/token-efficient-change-detection-llm-apis-561D/b3it_monitoring/assets/updated_fig5.pdf)) even detected a confirmed model *replacement*: the switch from Mistral-7B-Instruct-v0.3 to Ministral-3-14B-Instruct-2512, which is twice the size (confirmed by the API provider's [changelog](https://docs.together.ai/docs/changelog#jan-29)).
>
> ### Q3: Unperceivable changes
>
> This is a misunderstanding. In that paragraph, we considered what would go wrong *if we didn't study the problem in the LAN regime*. The problem was that the distributions would become indistinguishable. When we wrote "in this regime" line 143, we meant *in the LAN regime*. So we are not claiming that we need the changes to be unperceivable, on the contrary. We shall clarify this point in our final version.
>
> ### Q4: Definition of temperature
>
> Temperature is a key parameter in LLM inference. The last layer of the Transformer outputs a vector of logits (real numbers), one for each token in the vocabulary. These numbers are then converted into a probability distribution with a softmax operation:
>
> $$p_i = \frac{e^{z_i / T}}{\sum_{j=1}^{V} e^{z_j / T}}$$
>
> Where $T$ is the temperature. Modifying it allows to adjust the level of randomness of the outputs: a low temperature assigns a high probability mass to the tokens which had the highest logits. In the limit, when $T=0$, the softmax becomes an argmax:
>
> $$p_i = \mathbb{1}[i = \arg\max_j z_j]$$
>
> Therefore, for a given prompt at $T = 0$, we would expect to always sample the same output -- if we didn't account for tied top logits and hardware non-determinism, which are at the heart of our work.
>
> All LLM APIs that we are aware of allow setting the temperature to 0.

---

> > ### Author Rebuttal · Reviewer_kgpd · 2026-04-01
> >
> > Technical matters were correctly addressed. However, I still believe that the poor writing style and the overall organization cannot be compensated from materials in the appendix.

---

> > > ### Author Response · Authors · 2026-04-07
> > >
> > > We thank Reviewer kgpd for acknowledging technical objections as resolved by our rebuttal.
> > >
> > > We note that the remaining concern is exclusively about presentation. Per ICML guidelines, the reviewer's score of 2 (Reject) corresponds to "writing so poor that it is not possible to understand its key claims". We believe this threshold is not met: the other three reviewers demonstrated a clear understanding of our contributions, and their reviews all rated Presentation as "good".
> > >
> > > We respectfully disagree with the claim that "*the rationale leading statistical developments is missing (see for instance 116 - 121)*". We took care to make this rationale prominent throughout the analysis, with progressive motivation before each theorem, and explicit "Takeaway" paragraphs. We asked for concrete suggestions on how to improve the narrative, but did not receive further guidance.
> > >
> > > We remain open to any actionable feedback to improve the presentation.

---

### Official Review · Reviewer_cuVM · 2026-03-16

**Soundness:** 3
**Presentation:** 3
**Significance:** 3
**Originality:** 3
**Overall Recommendation:** 4
**Confidence:** 3

**Summary:**

This paper studies the problem of remote change detection in large language model (LLM) APIs, where users need to identify whether the behavior of a deployed model has changed over time without access to model internals. Existing approaches typically require either white-box access to model parameters or grey-box access to log-probabilities, while purely black-box methods often incur high query costs. To address this challenge, the authors propose Black-Box Border Input Tracking (B3IT), a token-efficient change detection framework that operates under a strict black-box setting, observing only the output tokens produced by the model. The key idea is to identify border inputs, i.e., inputs for which multiple output tokens have comparable probabilities. The authors theoretically argue that such inputs are particularly sensitive to model changes in the low-temperature regime, making them effective probes for detecting distribution shifts. Based on this insight, the proposed method tracks model outputs on selected border inputs and performs statistical tests to detect behavioral changes. The paper provides theoretical motivation linking border inputs to properties such as the Jacobian and Fisher information of the output distribution.

**Compliance With Llm Reviewing Policy:**

Affirmed.

**Final Justification:**

Thanks for the rebuttal. My concerns have been addressed.

**Key Questions For Authors:**

1. Border input discovery: How computationally expensive is the process of identifying border inputs in practice? Could the authors provide more details on how many queries or iterations are typically required to discover useful inputs?

2. Robustness to decoding settings: The analysis focuses on low-temperature regimes. How sensitive is the proposed method to different decoding parameters (e.g., temperature, top-k, or nucleus sampling) commonly used in API deployments?

3. Types of detectable changes: Could the authors clarify what categories of model changes are easiest or hardest to detect with B3IT (e.g., small alignment updates vs. large architecture changes)?

4. Generalization across models: Do border inputs discovered for one model remain effective probes when monitoring other models or versions of the same model?

5. Practical deployment: In a real monitoring system, how frequently should B3IT be run to balance detection latency and query cost?

Answers to these questions would help clarify the practical applicability and limitations of the proposed approach.

**Limitations:**

Yes

**Strengths And Weaknesses:**

# Strengths

1. Practical and well-motivated problem. The paper addresses an increasingly relevant problem: detecting behavioral changes in deployed LLM APIs. As models are frequently updated without explicit versioning or transparency, reliable monitoring mechanisms are valuable for downstream applications, safety evaluation, and benchmarking.

2. Strict black-box setting. A notable strength of the proposed approach is that it operates under a strict black-box assumption, observing only output tokens. This significantly improves the practicality of the method compared to prior approaches that rely on log-probabilities or model internals.

3. Token-efficient design. The proposed B3IT framework is designed to be query-efficient, which is important in real-world API settings where monitoring costs scale with the number of requests. Experimental results indicate that the method can detect model changes with significantly fewer queries than existing black-box approaches.

4. Intuitive and theoretically motivated idea. The concept of border inputs is intuitive and grounded in statistical reasoning. The authors connect the sensitivity of such inputs to properties like the Jacobian and Fisher information of the output distribution, providing theoretical intuition for why these inputs are effective probes for detecting changes.

5. Empirical validation. The paper includes both controlled (in-vitro) experiments and real-world (in-vivo) evaluations, which strengthens the empirical credibility of the proposed method. Comparisons with prior methods help illustrate the advantages of B3IT in terms of efficiency and detection capability.

# Weaknesses

1. Limited theoretical guarantees. While the paper provides theoretical intuition linking border inputs to statistical sensitivity, the analysis remains somewhat informal and limited to specific regimes (e.g., low-temperature settings). Stronger theoretical guarantees or clearer formalization of the detection performance would strengthen the contribution.

2. Dependence on border input discovery. The effectiveness of the method relies on identifying suitable border inputs. However, the process of discovering or constructing such inputs may itself require nontrivial exploration or assumptions about the model behavior. The paper could benefit from a more detailed discussion of the computational cost and robustness of this step.

3. Evaluation scope.  Although the experiments demonstrate promising results, the evaluation could be expanded in several directions:
(1) More diverse model families and API providers; (2) Additional types of model changes (e.g., safety tuning, instruction tuning, architecture-level updates);  (3) Longer-term monitoring scenarios. This would help better characterize the robustness of the method across realistic deployment conditions.

4. Practical deployment considerations. Some operational aspects remain unclear, such as how frequently monitoring should be performed, how to handle noisy outputs, and how sensitive the method is to stochastic decoding settings.

---

> ### Author Rebuttal · Authors · 2026-03-31
>
> We thank the reviewer for the valuable feedback, and for appreciating the operational setup and the theoretical guidance of our work. Regarding the weaknesses and questions mentioned in the review:
>
> ### W1: Limited theoretical guarantees, only low temperature
>
> We note that Theorem 3.1 and Lemma 3.2 hold for any temperature. These results, in addition to being novel by themselves, enable us to prove Theorem 3.3, showing that detectability is maximal for Border Inputs at low temperature. Low temperature is thus the optimal regime for change detection, not a condition for our main results to hold. As $T=0$ can be set for all LLM APIs we are aware of, this becomes the basis of the B3IT technique.
>
> ### W2: Cost and robustness of Border Input discovery
>
> At the end of page 7, we report that Border Input discovery costs 0.0045 dollars per successful endpoint on average. Figure 4 also shows how many queries are typically required to identify Border Inputs: for instance, at $T=0$, 50% of endpoints require less 300 requests (100 prompts) per successful BI.
>
> For more discussion on BI search strategies, please see our answer to mhWu (Q3). On the fraction of endpoints where we can easily identify BIs, please see our answer to Cccd (W1,W3).
>
> ### W3: Evaluation scope
>
> We validate B3IT extensively:
>
> * in-vitro on 9 LLMs from 7 model families, on 5 difficulty scales from the TinyChange benchmark (Figure 3, Figure 7);
> * in-vivo on 93 endpoints (64 models, 38 providers) for the BI prevalence study, and 54 endpoints for continuous monitoring.
>
> We now have results for longer-term monitoring (see our [updated Figure 5](https://anonymous.4open.science/r/token-efficient-change-detection-llm-apis-561D/b3it_monitoring/assets/updated_fig5.pdf)). Multiple undisclosed changes were identified. One of the changes was confirmed as a model swap from the provider's [changelog](https://docs.together.ai/docs/changelog#jan-29).
>
> ### W4: Practical deployment considerations
>
> We send queries every 24h, but determining a frequency is up to the auditor, depending on their budget.
>
> Inference non-determinism should already be covered by our in-vivo experiments. We are unsure what the reviewer refers to by "noisy outputs" beyond this, and would welcome clarification so we can address it properly.
>
> Regarding the stochastic decoding settings, please see our response to Q2 below.
>
> Our source code can also serve as an example of a practical deployment.
>
> ### Q1: Cost of Border Input Discovery
>
> Please see our answer to W2.
>
> ### Q2: Robustness to decoding settings
>
> Our method focuses on APIs where the temperature can be set by the user, which represents all LLM APIs that we are aware of. Decoding parameters such as top-k, top-p or min-p are also set by the user, so we believe that it would be outside of the scope of this study to be robust to these settings.
>
> ### Q3: Types of detectable changes
>
> Our evaluation on the TinyChange benchmark (Figures 3 and 7) show that detection is easier for larger changes, and harder for smaller changes. Regarding architectural changes, our in-vivo experiment detected the model swap from Mistral-7B-Instruct-v0.3 to Ministral-3-14B-Instruct-2512 unambiguously, with the mean TV distance jumping from $\approx 0.05$ to $1$ (see W3).
>
> ### Q4: Generalization across models
>
> This is a very interesting question. As we already had the data, we tested whether BIs found for an endpoint are more likely to be BIs for another endpoint.
>
> We ran the Cochran–Mantel–Haenszel test.
>
> | Pair type                 | Models | Endpoints | CMH Odds Ratio (95% CI) | *p*     |
> | ------------------------- | ------ | --------- | ----------------------- | ------- |
> | Same model, diff provider | 28     | 86        | 2.40 (2.12, 2.72)       | 1.4e-49 |
> | Different model           | 55     | 104       | 1.07 (1.04, 1.11)       | 1.2e-04 |
>
> BIs are indeed correlated for the same model on different providers: a BI on an endpoint is 2.4x more likely to be a BI on another same-model endpoint than the base rate. This correlation is expected since the model is supposed to be the same.
>
> For different models, the effect is much smaller and close to independence: a BI on a model is only 1.07x more likely to be a BI on a different model. The fact that BIs don't generalize across models relates to their effectiveness as change detectors: by design, they are very likely to break down on a model change, which also means they are unlikely to generalize across models.
>
> We will add these results to the paper.
>
> ### Q5: Practical deployment considerations
>
> The choice of how often to run B3IT will be up to the auditor, with their specific budget and requirements. Given its low cost, nothing prevents B3IT from being run much more often than we did (every 24h). As we request a single token of output, cost is fully predictable. The auditor may also use our Figure 1 to decide on the number of prompts and samples per prompt that will achieve the best performance given their budget.

---

> > ### Author Rebuttal · Reviewer_cuVM · 2026-04-03
> >
> > Thanks for the rebuttal. My concerns have been addressed.

---

> > > ### Author Response · Authors · 2026-04-04
> > >
> > > We thank Reviewer cuVM for acknowledging concerns as fully resolved. We are pleased that the long-term monitoring data and the Border Input generalization results adequately addressed the raised points; we will integrate these results into the final version of the paper.

---

### Official Review · Reviewer_Cccd · 2026-03-20

**Soundness:** 3
**Presentation:** 3
**Significance:** 2
**Originality:** 4
**Overall Recommendation:** 5
**Confidence:** 4

**Summary:**

The authors present a black-box LLM change detection method, called B3IT (black-box border input tracking), that requires only observed token distributions in the low (ideally zero) temperature regime. The authors provide non-asymptotic guarantees on type I and II errors, and show, via extensive experimentation across a large collection of open-weight models and closed-source APIs, that their method outperforms existing black-box methods and performs at the level of the best grey-box (i.e. access to log-probs) methods.

The authors' key insight is the following: using local asymptotic normality analysis, the power of an optimal test for detecting changes in an LLM's output token distribution depends on a signal-to-noise ratio term that either 1) converges to zero when the output token distribution is dominated by a single mode, or 2) diverges when there are two or more output tokens with maximum likelihood. Such inputs are called Border Inputs (BIs), and B3IT is constructed to take advantage of this phase transition in two stages:

1. Initialization. Given a minimal input length prompt, determine the set of BIs and their associated output token sets. Experiments are run in the following way: submit random minimal length prompts a total of m=3 times each at T=0 (or the lowest possible temperature) until at least 5 BIs are determined. BIs are determined whenever any of the sampled output tokens is different from the others.
1. Detection. Whenever a test is to be run, sample each of the 5 BIs m=3 times, looking for collapse to a single output token amongst the samples. Whenever this occurs, declare the LLM changed.

**Compliance With Llm Reviewing Policy:**

Affirmed.

**Final Justification:**

This is a sound and interesting paper, based in theory and reduced to practice. My concerns were primarily about applicability, but these were sufficiently addresses in the rebuttal. I recommend accepting.

**Key Questions For Authors:**

- What are some ideas for improving the method is that it can be applied to much larger number of models? Potential areas include BI detection and reasoning models.
- Why not run the same performance analysis for reasoning models, allowing for larger output tokens? Seems like a miss letting an arbitrary output token limit disqualify the assessment of B3IT on these important examples.
- Can any of the claims of the LAN analysis be theoretically extended beyond the weight perturbation case analyzed here? It's notable that none of the paper's experiments are actually pure examples of weight perturbations.

**Limitations:**

I do not think that the lack of applicability to a large number of models of interest is appropriately addressed.

**Strengths And Weaknesses:**

Strengths:

- Theorem 3.3 as well as the results leading to it (e.g. Equation 1 and Lemma 3.2) constitute novel technical insights that have not been demonstrated before for LLMs.
- B3IT is a practical black box change detection method based on these insights that is relatively inexpensive to run.
- The authors' provide extensive evaluation across many open-weight models and closed-weight APIs and show that the performance of B3IT in detecting changes is much better than current black-box methods and on par with current grey-box methods.
- The authors provide extensive, sound technical derivations and proofs of their theorems and claims.

Weaknesses:

- Despite the authors' claim that BIs are easy to discover, when T=0, B3IT found at least one BI in only 62% of the 93 commercial LLM APIs tested, and the percentage that found at least 5 is not reported. This suggests a general lack of applicability to the models of interest.
- Some LLMs cannot be used at T=0, and others operate differently at T=0 than the limit T -> 0 would suggest, again limiting applicability.
- It is stated that the method does not work for reasoning models since they rarely if ever return output tokens within short output token limits, and again limiting applicability.
- Key insights depend on LAN analysis, which is only valid for small changes in the weights. The analysis does not directly apply to large changes in the weights or architectural changes leading to a difference in the set of weights themselves. It should be noted that the authors do examine the performance of B3IT in some of these cases, e.g. AUC-ROC dependence on pruning fraction and LORA fine-tuning, and still find that the method is superior to other black-box methods.
- There are other sources of noise in the output token distribution that neither the analysis nor the experiment explicitly account for.

---

> ### Author Rebuttal · Authors · 2026-03-31
>
> Thank you for your review, valuable feedback and having appreciated the novel technical insight and practicality/performance of the approach.
>
> ### W1, W3: Method doesn't apply to all endpoints
>
> We showed that BIs could be found in the majority of endpoints with a very limited query budget (1,000 queries per endpoint, at an average cost of $0.0045 (0.45 cents) per endpoint), and the simplest possible search strategy.
>
> We re-ran our search on an updated list of 134 endpoints. We found the following results, showing that the applicability is superior to what our initial numbers may have suggested.
>
> |                | ≥ 1 BI        | ≥ 5 BIs      |
> |----------------|---------------|--------------|
> | 1k prompts     | 95/134 (71%)  | 86/134 (64%) |
> | 2k prompts     | 99/134 (74%)  | 90/134 (67%) |
> | 2k + reasoning | 102/134 (76%) | 91/134 (68%) |
>
> The jump from our initial 62% to 71% at 1k prompts comes from newer endpoints: at the time we had 93 endpoints, now 134, with 57 endpoints overlapping.
>
> We report fractions of endpoints with at least 1 and at least 5 BIs, per your point in W1. The former is an approximation of the latter if a 5x larger query budget was used.
>
> We implemented your suggestion to cover reasoning endpoints (increasing the cost by requesting the full reasoning traces, and focusing on their first token of output), yielding a small improvement in coverage.
>
> We also show that increasing the query budget (from 1k to 2k prompts) impacts the reported number (71% to 74% covered).
>
> ### W2: $T=0$ vs $T \to 0$
>
> Regarding LLMs not supporting $T=0$, such as chat interfaces, change detection for these interfaces would indeed be more difficult to perform, and is out of scope for our work. All LLM APIs that we are aware of allow setting the temperature to 0. We argue that the expectation of stability is also stronger on APIs (which are the typical interface for researchers, application developers and regulators) than on chat interfaces.
>
> We have indeed identified that some endpoints behave differently at $T=0$ than $T$ very close to $0$, e.g. $T=10^{-10}$. But this doesn't limit the applicability of the method. We performed our experiments using $T=0$, but we could have chosen a very small temperature instead and have covered more endpoints, and users of the method may elect to do so.
>
> ### W4: Analysis focuses on small weight changes
>
> The LAN framework is deliberately chosen for its tractability: to our knowledge, we are the first to identify that the change detection problem admits a rigorous analysis in this regime beyond simple multinomial testing (non-informative on relevant auditing design choices). For large perturbations or architectural changes, the non-convex loss landscape of LLMs renders the problem analytically intractable. Furthermore, the LAN regime identifies the precise quantities governing detection performance: the Fisher information of the output distribution, the Jacobian, and the temperature, and directly informs B3IT's design.
>
> Experimentally, B3IT significantly outperforms all black-box baselines well beyond the theoretical regime, across perturbations explicitly outside the LAN assumptions. For instance, random pruning up to 100% of the weights, up to 4,096 steps of fine-tuning (Figure 3, Figure 7), but also i.i.d. Gaussian weight perturbations across a wide range of magnitudes (Figure 7, top left), which directly instantiates the small-perturbation regime of the theory. Our updated in-vivo experiments (see our [updated Figure 5](https://anonymous.4open.science/r/token-efficient-change-detection-llm-apis-561D/b3it_monitoring/assets/updated_fig5.pdf)) also detected a confirmed model *replacement* from Together AI. The theory thus provides principled grounding for a method that generalizes empirically both within and beyond it.
>
> ### W5: Other sources of noise
>
> We are not sure about what other sources of noise are considered here, and we would be willing to engage during the discussion period to clarify this point. Note that our in-vivo experiments should account for all sources of noise, even ones we may not have thought about.
>
> ### Q1, Q2: Applying method to more models
>
> Please see our comment on W1 and W3 which should answer both of these questions.
>
> ### Q3: Extending the analysis
>
> We do have an experiment on the TinyChange "random noise" difficulty scale, which adds gaussian noise to all parameters, with $\sigma=2^{-30},2^{-29},...,2^{0}$. It is shown in Figure 7, top left (admittedly in the Appendix; Figure 3 in the main body shows the fine-tuning difficulty scale, and line 371 links to Figure 7).
>
> We also note that any change that preserves the architecture can be written as $\theta_0 + \epsilon h$ (with $h$ a unit direction and $\epsilon$ a magnitude).

---

> > ### Author Rebuttal · Reviewer_Cccd · 2026-03-31
> >
> > Thank you for the thoughtful rebuttal, I feel like my concerns were addressed, and I have raised my score accordingly.
> >
> > On the point about noise, I had in mind non-model-related sources of variation due to hardware, etc., but realize that the conclusion includes addressing these as future work under the title of inference non-determinism, and so am ok to not pursue this further.

---

> > > ### Author Response · Authors · 2026-04-04
> > >
> > > We thank Reviewer Cccd for the score increase. We are glad the additional results addressed the concerns on applicability. We will incorporate these new findings and the updated coverage analysis into the final version of the paper.

---

### Decision · Program_Chairs · 2026-04-30

**Decision:**

Accept (regular)

**Comment:**

The paper studies strict black-box change detection for LLM APIs. It argues that border inputs become highly sensitive at low temperature, then turns that insight into B3IT, a two-stage detector that first finds border inputs and then checks support mismatch. The paper also evaluates the method in vitro on TinyChange and in vivo on commercial APIs. The result is a mix of new theory and a usable system, which is why the paper is more than a narrow application note.

The theory leads to a concrete algorithm with low query cost. The empirical story is also strong, with clear gains over black-box baselines and close performance to the grey-box LT baseline. The rebuttal added useful coverage numbers for reasoning endpoints and longer monitoring, which makes the practical case more credible. Several reviews converged on the same point: the paper is doing real work, not just repackaging existing black-box methods.

The rebuttal moved the main reviews in the right direction & the authors added larger coverage numbers, a reasoning-endpoint variant, and longer monitoring results. That makes the practical story more credible, and it also makes the limits of the method clearer.

The natural audience for this paper is the crowd that works on API auditing, deployed-model monitoring, and statistical testing for black-box systems. They will read this paper for the border-input phase transition and for a usable detector with real cost savings.